



**In-situ Rb-Sr geochronology of white mica in young metamafic and**
**metasomatic rocks from Syros: testing the limits of LA-ICP-MS/MS mica**
**dating using different anchoring approaches**
Jesús Muñoz-Montecinos [1], Andrea Giuliani [2], Senan Oesch[2], Silvia Volante [1], Bradley
Peters[2] , Whitney Behr [1]
[1] Institute of Geology, Department of Earth Sciences, ETH Zurich
[2] Institute of Geochemistry and Petrology, Department of Earth Sciences, ETH Zurich
* Correspondence to: Jesús Muñoz-Montecinos (jmunoz@ethz.ch)
**Abstract**
The recent development of LA-ICP-MS/MS has revolutionized Rb-Sr mica dating allowing to
obtain isotopic data within their microstructural context. While effective for old and felsic
materials, this method presents challenges for young metamafic and metasomatic rocks due to
limited radiogenic ingrowth associated with low Rb/Sr and young ages. We quantitatively
address these limitations by combining laser ablation ICP-MS/MS and MC-ICP-MS data for
coexisiting white mica and epidote, respectively, for 10 Cenozoic metamorphic rocks from
Syros Island (Greece). White mica analyses from metamafic and metasomatic rocks yield
limited Rb/Sr spread, which typically does not exceed one order of magnitude ($^{87}$Rb/$^{86}$Sr = 14
to 231 for the combined dataset), and low radiogenic $^{87}$Sr/$^{86}$Sr (generally <0.8), resulting in
high age uncertainties of typically 10 to 50% RSE, and thus hampering robust geological
interpretations. Epidote $^{87}$Sr/$^{86}$Sr values range between ~0.705 and 0.708. The former (lower-
end) is expected for typical, unaltered metamafic materials, whereas the latter is interpreted to
reflect fluid-rock interaction along shear zones, with fluids derived from or having interacted
with  more radiogenic lithologies. These atypical values suggest that a commonly assumed
value of 0.703 for mafic rocks may not always be representative. Anchoring white mica Rb/Sr
to epidote $^{87}$Sr/$^{86}$Sr data improves age accuracy and precision substantially (e.g., 29 ± 17 Ma
vs 47.2 ± 4.4 Ma for sample SYGR36). The new ages obtained in this study are consistent with
multiple events previously recorded in Syros and the Cyclades blueschists unit including: i)
metasomatism at near-peak to epidote blueschist-facies during early exhumation (47.2 ± 3.8
Ma to 41.1 ± 3.1); ii) a late stage of high-pressure exhumation and metasomatism transitioning
to blueschist-greenschist-facies (20.8 ± 3.1 Ma to 19.8 ± 5.2 Ma). Anchored white mica Rb/Sr
dates thus allow us to discriminate events of fluid-rock interactions and metasomatism
associated with shear zone deformation at the subduction interface.
**Keywords**
Rb-Sr dating; phengite; epidote; Syros; metamorphism; metasomatism



# Introduction

Subduction zones host a wide range of mechanical and chemical processes that occur at various spatial and temporal scales including but not limited to seismicity (e.g., Muñoz-Montecinos et al., 2021; Wirth et al., 2022), element and nutrient recycling/transfer (e.g., Li et al. 2021, Tumiati et al., 2022; Rubatto et la., 2023), volcanism (Breeding et al., 2004) and orogenesis (e.g., Burg and Bouilhol, 2019). These processes are temporally associated with metamorphism and metasomatic events of rocks conforming the deep subduction zone, and occur at time-scales ranging from steady-state tectonics (e.g., foliation development and nappe stacking over millions of years) to nearly instantaneous mineral growth and fluid flow (John et al., 2012). Constraining the timing of high-pressure and low-temperature (HP-LT) crystal growth and fabric development within metamafic lithologies that once occupied the subduction interface is therefore crucial for understanding deep tectono-thermal processes occurring at depth. However, geochronological studies of these HP-LT rocks are commonly challenged by the difficulty of precisely determining the age of subduction-related metamorphic events.

While the U-Pb system has been conventionally employed to date metamorphic, magmatic and hydrothermal events, it relies on the presence of U-bearing accessory phases such as zircon, monazite, titanite, rutile, and apatite which may be scarce, too small to be targeted, or absent in high-pressure metamafic rocks (e.g., Timmermann et al., 2004; Engi et al. 2017; Holtmann et al., 2022; Bastias et al., 2023: Volante et al. 2024 and reference therein). Moreover, mid- to low-temperature metamorphic and metasomatic events along the burial-to-exhumation path are commonly not traceable using U-Pb geochronology due to the higher closure temperature of most geochronometers (e.g., Chew and Spikings, 2015). Another common problem and challenging task when using U-Pb of accessory minerals to date mafic rocks is combining their textures with fabric development, thus hindering the chronological link of these U-bearing phases to a particular microstructure. In such cases, exploring alternative minerals and systematics becomes crucial to obtaining a more comprehensive and accurate record of the deformation and metamorphic history.

White mica is a common mineral in metamafic, HP-LT lithologies that is stable throughout prograde and retrograde reactions (Schmidt et al., 2004; Halama et al., 2020). Enrichment in Rb makes white mica a suitable Rb-Sr geochronometer for dating subduction-related metamorphic processes, also considering the high closure temperature of the Rb-Sr system at static, fluid-absent conditions (500-600 °C – von Blanckenburg et al, 1989; Villa, 1998; Glodny et al. 1998, 2008). While multimineral Rb-Sr internal isochron analyses of subduction zone rocks have been extensively utilized, often yielding robust ages (Glodny et al., 2004, 2008; Wawrzenitz et al., 2006; Bröcker et al., 2013; Kirchner et al., 2016; Angiboust and Glodny, 2020), significant challenges remain to be addressed. These include: i) Sr isotope disequilibrium between micas and the other mineral phases, which are commonly included in Rb-Sr isochronous arrays; ii) post-deformation, low-temperature magmatic alteration or fluid-assisted recrystallization, which might affect pristine Rb-Sr isotope compositions; iii) thermally-induced diffusion processes that can also impact the Rb-Sr record (Glodny and Ring





2022); and iv) potential inheritance within mica grains or across mica populations (Villa, 2016;
Barnes et al., 2024). Therefore, grain size, deformation, alteration and fluid availability might
control Rb-Sr isotope variability. For example, Glodny et al. (2008) showed that large crystals
of biotite partially preserved the Grenvillian Sr-isotopic composition related to granulite-facies
metamorphism, whereas submillimeter-sized biotite in fully re-equilibrated eclogite rocks
yielded a different Sr-isotopic signature due to an overprinting Caledonian eclogite-facies
event. These variations in mica Rb-Sr systematics, and the processes underpinning them, can
be addressed directly using laser ablation methods.

Although substantially less precise, in-situ Rb-Sr dating of white mica using a triple quadrupole
inductively coupled plasma mass spectrometer associated with a laser ablation system (LA-
ICP-MS/MS) offers significant advantages over conventional ID-TIMS methods. This in-situ
method eliminates the need for mineral separation and time-consuming chromatographic
column chemistry, enabling quick, cost-effective analyses. It further allows one to constrain
potential zoning in Rb-Sr isotope distribution (Kutzschbach and Glodny, 2024; Rösel and Zack,
2021) and to link multiple grain populations to specific microstructural domains, hence
preserving essential textural information which are otherwise inaccessible. Thus, potential age
variations among different white mica populations (e.g., syn- to post-kinematic grains) within
distinct microstructural domains such as microfolds, shear bands, and boudin necks permits a
more accurate interpretation of ages as shown in previous mica Rb/Sr studies of metamorphic
rocks by LA-ICP-MS/MS (Gou et al., 2022; Gyomlai et al.,2022, 2023a; Kirkland et al., 2023;
Ribeiro et al., 2023; Ceccato et al., 2024; Barnes et al., 2024).

In-situ Rb-Sr geochronology has been increasingly utilized to constrain the timing of
deformation events along Precambrian shear zones in felsic igneous rocks (e.g., Olierook et al.,
2020; Wang et al., 2022; Ribeiro et al., 2023). For instance,  Tillberg et al. (2021) targeted
white mica and other potassic mineral phases (e.g., illite) to constrain distinct populations of
Precambrian and Paleozoic brittle veins and ductile shear zones. Although age (sub-)clusters
were challenging to distinguish, this method showed great potential for dating multiple events
of fault activation and reactivation (Kirkland et al., 2023). Mica Rb/Sr studies on mafic
lithologies are comparatively limited. A recent study from Gyomlai et al. (2023a) on mafic
blueschist from the Kampos belt on Syros island (Greece) presented data on the timing of fluid-
rock interactions along the subduction interface. However, their large uncertainties precluded
the distinction between peak-pressure metamorphism, retrogression and/or partial
recrystallisation of white mica under blueschist- to greenschist-facies conditions. More
accurate age constraints were obtained by Barnes et al. (2024) for a small set (n = 4) of samples
from the Syros Island, where mica Rb/Sr dating was combined with initial Sr isotope
constraints provided by epidote and apatite.

These studies and the increasingly widespread application of mica Rb-Sr dating by LA-ICP-
MS/MS to date magmatic and hydrothermal events (e.g., Redaa et al., 2022; Wang et al., 2022;
Zametzer et al., 2022; Huang et al., 2023; Giuliani et al., 2024) highlights the versatility of in-
situ mica Rb-Sr geochronology to investigate different rock-types and  geological questions.
However, lithologies with low Rb contents (e.g., < 30 ppm in mafic rocks) and associated low



Rb/Sr micas result in low accuracies of mica Rb/Sr ages, even where hundreds of analytical
spots were measured (Tillberg et al., 2020). This limitation is exacerbated by dating micas in
young (i.e., Cenozoic) metamafic rocks where ingrowth of radiogenic Sr is limited.
In this contribution we aim to address the current limitations of in-situ Rb-Sr dating of white
mica in young metamafic and metasomatic rocks that typically contain mica crystals with low
Rb/Sr, and provide strategies to obtain robust Rb-Sr ages from these lithologies using laser
ablation methods. We present new data from 10 samples from Syros Island (Kampos Belt and
Megas Gialos area; Greece). We present new petrographic, textural and microstructural
analysis of greenschist- to eclogite/blueschist-facies rocks combined with laser ablation Rb/Sr
analyses of white mica and laser ablation multi-collector (MC) ICP-MS Sr isotope analyses of
epidote for 10 samples from the Syros Island (Kampos Belt and Megas Gialos Beach; Greece),
complemented with bulk rock Sr isotopes for some of the samples. Although the general
architecture and structural relationships of blueschist- to eclogite-facies rocks in Syros are still
debated (e.g., Keiter et al., 2011; Laurent et al., 2018; Kotowski et al., 2022), the subdivision
of geological units, P-T conditions and the timing of metamorphic burial and exhumation are
well-constrained, making Syros an ideal case study for our purpose. We demonstrate that in
these young (Cenozoic) metasomatic and metamafic rocks, low Rb/Sr ratios commonly
preclude precise dating of mica by LA-ICP-MS/MS. Anchoring the mica-based Rb-Sr isochron
to an initial (or 'common') $^{87}Sr/^{86}Sr$ provided by either a low-Rb/Sr cogenetic phase such as
epidote or a geologically meaningful 'model' (e.g., Rosel and Zack, 2021) helps circumvent
this problem.

GEOCHRONOLOGY
Discussions
EGU

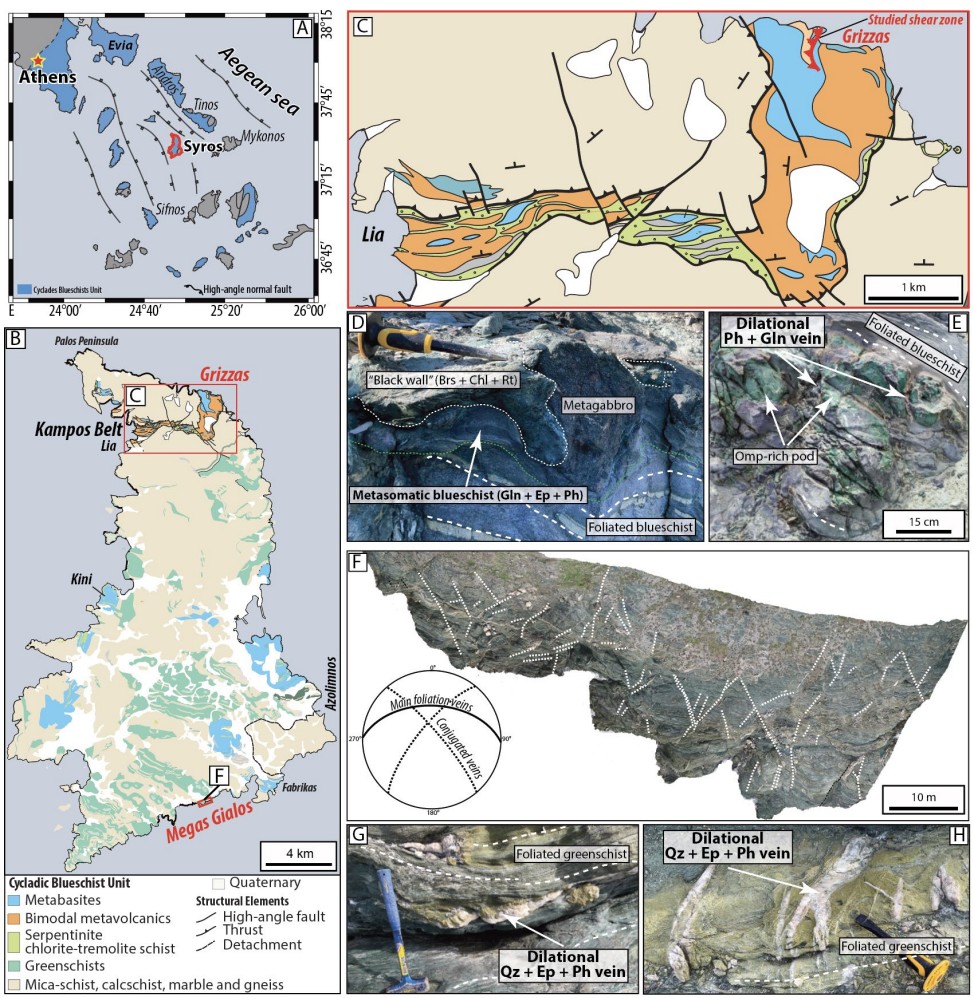

*Figure 1*. *A. Simplified geologic map of the Cyclades highlighting the location of Syros Island. B. Simplified*
*geologic map of Syros Island (modified from Keiter et al., 2011); the study localities are highlighted in red. C.*
*Zoom in of the Kampos Belt; the study locality is shown in red. D. Field image of an omphacitite pod embedded*
*in a foliated blueschist matrix within the Grizzas shear zone, the former contains some of the studied glaucophane*
*+ phengite dilational veins (sample SYGR41). E. Representative field image of metasomatic rocks from the*
*Grizzas shear zone; note the occurrence of smeared metagabbro blocks surrounded by metasomatic rinds and*
*black walls within a foliated blueschist and a chlorite-tremolite schists matrix. F. Orthomosaic image from the*
*Megas Gialos outcrop (inset from panel B; modified from Muñoz-Montecinos and Behr, 2023). The lower-*
*hemisphere stereoplot depicts the orientation of the veins following the main foliation as well as those sets oblique*
*to it (dashed white lines illustrate the orientation of conjugated vein sets), represented here by samples SYMG08.3*
*and SYMG02, respectively. G and H. Examples of dilational veins containing epidote fibers along with phengite.*



# Geological setting

## Syros Island

The HP-LT (high-pressure, low-temperature) rocks from Syros Island belong to the Cyclades Blueschists Unit (CBU) cropping out along the Aegean Sea (**Figure 1A and 1B**). The CBU is interpreted to represent exhumed fragments of the subducted Adriatic plate and HP-LT meta-ophiolites of a northward-dipping subduction event between the Eurasian and African plates (Gautier and Brun, 1994; Jolivet et al., 2010; Soukis and Stockli, 2012). The CBU is subdivided into three subgroups (Glodny and Ring, 2022), from which the Top and Middle CBU nappes are relevant for this study. The Top CBU nappe crops out in Syros as a narrow belt, known in the literature as the Kampos Belt, to which one of the study localities belong to: the Grizzas shear zone (**Figure 1C**). It is composed of abundant metavolcanic materials with a bimodal composition (mafic and felsic) along with metagabbros, serpentinites, tremolite-chlorite, talc- and garnet schists (Keiter et al., 2011). The Kampos Belt lithologies reached peak blueschist- to eclogite-facies conditions of 480–560°C and 1.6–2.2 GPa (e.g., Trotet et al., 2001; Laurent et al., 2018; Cisneros et al., 2020). The Middle CBU nappe is the most abundant unit and is mainly composed of a relatively coherent intercalation of marbles, metasediments and metabasites (**Figure 1B**). This latter lithotype represents the studied lithology at the Megas Gialos locality (**Figure 1B and 1F**), displaying a pervasive exhumation overprint transitioning from blueschist- to greenschist-facies from 450 to 400 °C and 1.4 to 1.0 GPa (Cisneros et la., 2020). These retrograde metamorphic conditions are associated with transient brittle fracturing and dilational veining (Muñoz-Montecinos and Behr, 2023), from which the investigated samples from Megas Gialos were collected.

The pre-subduction architecture of the CBU resulted from Triassic rifting of the basement accompanied by deposition of passive margin sediments and carbonates (Keay, 1998; Seman et al., 2017). Rifting occurred at c. 80 Ma, thinning the lithosphere and producing small-scale oceanic basins along with passive margin depocenters (Keiter et al., 2011; Cooperdock et al., 2018; Kotowski et al., 2022). In the Kampos Belt (Gryzzas locality), U-Th-Pb SHRIMP zircon analyses in a metagabbro and a meta-plagiogranitic dike reveal two age populations, one at c. 80 Ma and a second one at 52.4 ± 0.8 Ma (Tomaschek et al., 2003). The older age likely reflects the magmatic crystallization, whereas the younger one dates the HP-LT peak metamorphism. Phengite and multi-mineral Rb-Sr (e.g., white mica + epidote + glaucophane +/- omphacite +/- garnet), phengite Ar-Ar and garnet Lu-Hf ages (mostly from the Lia side, hereafter referred to as the Western Kampos Belt) are in the range of 55 to 44 Ma, and were interpreted to reflect the timing of prograde-to-peak HP-LT metamorphism (see Kotowski et al., 2022 and references therein). The initial stage of exhumation under blueschist-facies conditions likely began at c. 44 Ma, and transitioned to greenschist-facies conditions between 34 and 20 Ma based on Ar-Ar and Rb-Sr multi-mineral (e.g., white mica + epidote + albite) geochronology (e.g. Putlitz et al., 2005; Uunk et al., 2018; Glodny and Ring, 2022 and references therein). Gyomlai et al. (2023a) obtained in-situ mica Rb-Sr ages from a single, c. 2 m-thick outcrop in



the range of 52.5 ± 11.6 to 12 ± 3.1 Ma, inferred to date metasomatism of metamafic rocks
during HP and fluid-rock interaction during late exhumation, respectively. Multi-mineral and
in-situ white mica Rb-Sr and Ar-Ar dating in the Middle CBU nappe yielded peak HP-LT
metamorphism ages of 45 to 37 Ma, whereas the pervasive blueschist- to greenschist-facies
metamorphism is dated at c. 39 to 19 Ma (Glodny and Ring 2022; Barnes et al., 2024; Kotowski
et al., 2022 and references therein).

# Samples and Petrography

In this section, we present key petrographic observations of the 10 samples from the Syros
Island that have been selected for Rb-Sr dating (**Table 1**), emphasizing the textural context of
white mica and epidote. Two additional samples (SYGR50 and SYGR44) have been analyzed
for epidote $^{87}Sr/^{86}Sr$ only. The investigated samples were carefully selected in order to
constrain the timing of fluid-rock interactions (metasomatism and veining) and to evaluate the
significance of $^{87}Sr/^{86}Sr$ isotopic values for anchoring white mica Rb-Sr isochrons. We targeted
our samples based on the presence of white mica in apparent textural equilibrium with epidote
(where present) and, for the Grizzas samples, the apparent absence of greenschist-facies
overprinting.
The samples coded SYGR ($N_{dated}$ = 7; $N_{total}$ = 9) all belong to the Grizzas locality in the
easternmost part of the Kampos Belt (**Figure 1B and 1C**). These samples were collected along
a north-dipping shear zone (hereafter referred to as the Grizzas shear zone), which juxtaposes
a massive to variably strained white metagabbro and blueschist-facies igneous breccia,
representing a region of high and localized strain (**Figure 1C;** see also Keiter et al., 2011).
Samples SYGR36 and SYGR44 correspond to relict (partially digested) blueschist blocks,
while sample SYGR50 represents a pristine, low-strain metagabbro. SYGR37 and SYGR38
represent the metasomatized mafic matrix wrapping around the metagabbro and blueschist
blocks (i.e., metasomatic rinds in **Figure 1D**), whereas sample SYGR42 is an altered
metagabbro (see **Table 1** for a summary of the studied samples). For comparison, a
metasedimentary rock sample (SYGR45) from a c. 70 cm thick discrete layer within the shear
zone as well as a felsic pod (sample SYGR58) contained within a moderately-strained meta-
igneous breccia (e.g., Keiter et al., 2011), were also targeted for dating. We emphasize the
occurrence of dilational phengite + glaucophane veins (such as sample SYGR41) cross-cutting
omphacitite pods (**Figure 1E**). The samples coded SYMG were collected from a retrograde
greenschist-to-blueschist-facies sliver located in the Megas Gialos locality (**Figure 1F**). The
selected vein samples SYMG02 and SYMG08.3 (**Figure 1G and 1H**) formed as dilational
fractures related to the ascent of deep subduction zone fluids towards the base of the fore arc
during the latest stages of HP-LT exhumation and extension (e.g., Muñoz-Montecinos and
Behr, 2023). Sample SYMG07 represents the greenschist host rock associated with the vein
samples SYMG02 and SYMG08.3



| Table 1. Sample summary | | | |
|---|---|---|---|
| Sample ID | Rock type and general structure | Mineral assemblage | Analysed microdomain |
| *Grizzas shear zone (NE Syros)* | | | |
| *Blueschist-facies* | | | |
| SYGR36 | Strongly foliated blueschist block | Gln + Ep + Wm + Gte + Omp + Rt | Wm defining the main foliation and pressure shadows around Gte |
| SYGR37 | Moderate to strongly foliated metasomatic rind | Gln + Ep + Wm + Chl | Wm defining the main foliation |
| SYGR38 | Weakly foliated metasomatic rind | Gln + Ep + Wm + Chl | Randomly oriented and interlocked Wm and Ep |
| SYGR41 | Dilational vein | Gln + Wm | Randomly oriented laths of Wm |
| SYGR42 | Moderately foliated metagabbro | Gln + Wnc + Omp + Ep + Wm + Rt | Shear bands defining the main foliation |
| SYGR44 | Moderately foliated blueschist block | Gln + Lws (now Ep + Wm) + Wm + Gte + Rt | Ep replacing Lws pseudomorphs |
| SYGR45 | Foliated metasediment | Wm + Gln + Gte + Ep + Tur | Wm aligned and oblique according to the main foliation |
| SYGR50 | Weakly to moderately foliated metagabbro | Omp + Ep + Gln + Wm | Ep defining the main foliation and within boudin necks |
| SYGR58 | Moderately foliated felsic pod | Qz + Wm | Wm aligned and oblique according to the main foliation |
| *Megas Gialos (SE Syros)* | | | |
| *(HP)Greenschist-facies* | | | |
| SYMG02 | Dilational vein | Qz + Ep + Wm + Ab | Ep fibers and Wm laths in close contact |
| SYMGG07 | Moderatly to strongly foliated | Ep + Ab + Chl + Act + Wm + Ttn | Ep and Wm defining the main foliation |
| SYMG08.3 | Dilational vein | Qz + Ep + Wm | Ep fibers and Wm laths in close contact |


## Blueschists, blueschist-facies metagabbro and greenschist


Samples SYGR36 and SYGR44 are relict blueschist blocks within a metasomatized sheared
matrix. Glaucophane, together with white mica and epidote define the penetrative foliation.
Texturally, white mica occurs as medium-grained laths and displays no evidence of kinks,
undulose extinction or mica fish. In sample SYGR36, white mica also occurs within pressure
shadows **(Figure 2A)** and boudin necks around garnet as well as oblique to the main foliation.
No significant chemical zoning patterns were observed **(Supplementary Figure S1A)**. Mostly,
white mica crystals defining the main foliation as well as those spatially related to pressure
shadows were targeted for dating. Sample SYGR44 texturally preserves lozenge-shaped
lawsonite pseudomorphs now composed of strain-free epidote (targeted for $^{87}Sr/^{86}Sr$ analyses)
and white mica **(Supplementary Figure S1G)**.

Sample SYGR50 is a low-strain white metagabbro composed of coarse-grained clinopyroxene
pseudomorphs (now glaucophane, winchite and omphacite) in a matrix of epidote. The (weak)
foliation is defined by elongated tabular crystals of epidote and subordinate white mica. Boudin
necks within large porphyroclasts are filled by epidote, white mica and garnet **(Supplementary**
**Figure S1H).** In this sample, epidote crystals defining the foliation and filling the boudin necks





were targeted for $^{87}Sr/^{86}Sr$ analyses. Overall, this sample represents the weakly-metasomatized
analogue of the altered metagabbro sample SYGR42.

Sample SYMG07 is a coarse-grained greenschist and represents the host rock associated with
the vein samples SYMG02 and SYMG08.3. The main foliation is defined by amphibole and
epidote, oriented laths of chlorite and white mica as well as stretched albite (**Figure 2B**).
Phengite grains from the matrix display weak core-mantle zoning patterns noticeable in back-
scattered electron imaging (**Supplementary Figure S1B**). The core of large white mica
crystals were targeted for dating, while the foliated matrix epidote was targeted for $^{87}Sr/^{86}Sr$
determinations, since these are interpreted as part of an equilibrium assemblage.

## Metasomatic rinds, metasomatized metagabbro and veins

Samples SYGR37 and SYGR38 represent the matrix wrapping around metagabbro and
blueschist blocks. These samples are coarse-grained, foliated schists composed mainly of
glaucophane, epidote, phengite and chlorite. White mica from both metasomatic rinds are
medium to coarse-grained and occur in sharp contact with glaucophane and epidote, displaying
no significant chemical zoning patterns were observed nor textural evidence of recrystallization
(**Figures 2C and 2D)**. Sharp contacts between white mica and epidote suggest textural
equilibrium between them (**Supplementary Figure S1C**). Thus, we targeted these
microdomains for white mica dating and $^{87}Sr/^{86}Sr$ determinations.

Sample SYGR42 is an altered metagabbro composed of porphyroclasts of Na-Ca amphibole
and omphacite after igneous clinopyroxene in a matrix of epidote and glaucophane (**Figure
2E**). Two generations of epidote, spatially associated with two distinct microdomains, are
observed. The first epidote generation grew as fine-grained, now heavily smeared crystals
occupying the interstitial matrix between porphyroclasts. This texture likely reflects epidote
growth after igneous plagioclase and subsequent deformation. The second epidote generation
grew in microdomains where a discontinuous foliation composed of tabular glaucophane and
epidote in sharp contact with white mica, wrapped around porphyroclasts and the fine-grained
epidote matrix. Within this second microdomain, white mica is medium- to coarse-grained and
displays evidence of recrystallization and subgrains. For this reason, coarse-grained white mica
crystals displaying no textural evidence for recrystallization, such as subgrains, kinks and
undulose extinction, were carefully selected for dating, whereas euhedral and tabular epidote
crystals in sharp contact with white mica crystals were targeted for $^{87}Sr/^{86}Sr$ analysis.

Sample SYGR41 is a glaucophane + white mica dilatational vein cross-cutting an omphacitite
pod. These veins display up to centimeter-sized and randomly oriented laths of white mica
(**Figure 2F**) displaying no evidence of deformation nor chemical zoning (**Supplementary
Figure S1D)**.

Samples SYMG02 and SYMG08.3 are dilatational veins crosscutting the foliated greenschist
hosts. Elongated epidote crystals occur spatially associated with white mica in sharp contact
suggesting contemporaneous precipitation from a fluid phase (**Figures 2G and 2H**). White



mica occur as euhedral, hundreds of μm long laths and correspond to strain free crystals with
no to faint chemical zoning (**Supplementary Figures S1E and S1F**). Thus, the most coarse
and pristine (e.g., unfractured) crystals were selected for white mica dating and epidote
$^{87}Sr/^{86}Sr$ analyses.

## Metasedimentary rock and felsic pod

Sample SYGR45 is a well foliated garnet, glaucophane, tourmaline, mica schist with minor
epidote (**Figure 2I**). Texturally, the foliated white mica generation is apparently overgrown by
a second, static generation characterized by laths oriented oblique to the main foliation (**Figure
2E**). To avoid potentially retrograde rims, cores of large crystals defining the pervasive
foliation and those of crystals oblique to it were targeted for dating. However, the resulting
ages for these two white mica generations were indistinguishable within uncertainty, therefore
the final age for this sample was calculated by clustering both datasets (see below).
Sample SYGR58 is a felsic pod contained within the blueschist-facies meta-igneous breccia.
They are composed mostly of quartz and phengite and subordinate epidote and garnet, the latter
typically replaced by chlorite. A first white mica generation defines the foliation, whereas a
second generation of laths are oriented oblique to it (**Figure 2J**). Although the two white mica
generations were separately targeted for dating, the resulting ages overlap and were merged for
the final sample age calculation (see below).

GEOCHRONOLOGY
Discussions

EGU



***Figure 2**. Photomicrographs (crossed polars) of the dated samples. A. General overview of the blueschist block*
*sample SYGR36 emphasizing the distribution of white mica along the foliation and typically around garnets*
*forming pressure shadows. B. General fabric of the greenschist sample SYMG07 displaying the association*
*between foliated epidote and white mica. Due to the significant amount of inclusions within epidote, only the*
*inclusion-free regions were targeted for laser ablation MC-ICP-MS analysis. C. Contact between altered*
*blueschist and metasomatic rind in sample SYGR38; note the relatively curvy-sharp contact between these two*





*domains as well as the relatively larger abundance of coarse grained white mica in the latter. D. Contact between*
*altered  blueschist and metasomatic rind in sample SYGR37. In this case, the contact is moderately- to highly-*
*strained resulting in a more diffuse appearance. E. Metasomatized metagabbro sample SYGR42 displaying*
*clinopyroxene pseudomorph porphyroclasts (now replaced by amphibole) in a foliated matrix composed of white*
*mica, glaucophane and epidote.  F. Dilational vein cross-cutting an omphacitite pod (sample SYGR41) with*
*strain-free, millimeter-sized white mica crystals in association with glaucophane.G. Dilational white mica +*
*epidote + quartz vein (sample SYMG08.3) with a texture characterized by epidote fibers and white mica laths. H.*
*Dilational white mica + epidote + quartz vein (sample SYMG02) displaying coarse-grained epidote in sharp*
*contact with finer-grained white mica. I. Metasedimentary rock sample SYGR45 highlighting white mica crystals*
*oriented parallel and oblique (static) to the main foliation as well as developing pressure shadows around garnet.*
*J. Felsic pod sample SYGR58 highlighting the distribution of white mica along the main foliation as well as some*
*grains oriented oblique to it in a matrix of quartz. Mineral abbreviations are from Whitney and Evans (2010).*
*Chl – chlorite; Ep – epidote; Gln – glaucophane; Grt – garnet; Omp – omphacite; Qz – quartz; Ttn – titanite;*
*Tur – tourmaline; Wm – white mica; Wnc – winchite.*

# Methods

## Laser ablation MC-ICP-MS

In-situ Sr isotope analyses of epidote were undertaken in two separate sessions (March 2023
and February 2024) using an ASI RESOlution 193 nm excimer laser ablation system interfaced
to a Nu Plasma II MC-ICP-MS at ETH Zürich and following a similar approach to that of
Fitzpayne et al. (2023) and Pimenta Silva et al. (2023). Analytical conditions included 80-100
μm spot size, a repetition rate of 5 (Mar-23) and 10 Hz (Feb-24), and laser fluence of ~4.0
(Mar-23) and 2.5 J/cm$^2$ (Feb-24). Each analysis consisted of a sequence of 40 seconds of
ablation and 45 seconds of washout and gas blank measurement. Total Sr signals varied widely
from ~1 to 15 V depending on the sample (**Supplementary Table S1**). Data reduction,
including corrections for isobaric interferences (Kr, Ca dimers, Ca argides, Rb) and
instrumental mass bias was performed using Iolite 4 (Paton et al., 2007, Paton et al., 2011).
Instrumental drift was evaluated by repeated measurement of clinopyroxene BB-1 (Neumann
et al., 2004; Fitzpayne et al., 2020), which was ablated every block of 15 unknowns including
secondary clinopyroxene standards (JJG1414; YY09-04; YY09-47; YY12-01) from Zhao et
al. (2020) (results included in **Supplementary Table S1**). All the data are reported relative to
BB-1 of $^{87}Sr/^{86}Sr$ of 0.704468 (Fitzpayne et al., 2020) via standard bracketing. $^{84}Sr/^{86}Sr$ of
clinopyroxene standards and epidote unknowns are generally within uncertainty of the natural
ratio (~0.0565). $^{87}Rb/^{86}Sr$ ratios are negligible (typically <0.001), which makes corrections for
$^{87}Sr$ ingrowth insignificant. Therefore, the reported Sr isotope ratios are considered to be equal
to the initial Sr isotope ratios at time of epidote crystallization.

## Laser ablation ICP-MS/MS

In-situ Rb-Sr isotope analyses of white mica in thin section were undertaken during two
sessions (October 2022 and May 2023) using an ASI RESOlution 193 nm excimer laser probe
interfaced to an Agilent 8800 ICP-MS/MS at ETH Zürich following the procedure outlined in
Giuliani et al. (2024) and Ceccato et al. (2024). The mass spectrometer was first
tuned in single-quad mode (i.e. no gas in the collision cell) to



optimize the Rb and Sr signals by ablating NIST612. Oxide production rate based on measurement of ThO/Th in NIST612 was ≤0.2 wt.%. After introducing ultrapure $N_2O$ gas (>99.99%) in the reaction cell (flow rate of 0.23-0.25 mL/min), a second tuning step was undertaken by ablating NIST610 to maximize production of $SrO^+$ ions while maintaining high sensitivity for $Rb^+$ ions. Interaction of $Sr^+$ ions with $N_2O$ resulted in conversion of ~89% of $Sr^+$ ions to $SrO^+$ based on monitoring of masses 88 ($Sr^+$), 104 ($SrO^+$) and 105 ($SrOH^+$). No detectable $RbO^+$ was recorded. Analytical conditions for mica analyses included 80-100 µm spot size, a pulse rate of 5 Hz, and laser fluence of ~3.5-4.0 $J/cm^2$. Each analysis consisted of a sequence of 40 seconds of ablation and 45 seconds of sample washout and gas blank measurement. Dwell times were of 100 ms for $^{85}Rb$, $^{86}Sr^{16}O$ and $^{87}Sr^{16}O$, 50 ms for $^{86}Sr$ and $^{87}(Sr+Rb)$, 20 ms for $^{88}Sr$, $^{88}Sr^{16}O$ and $^{88}Sr^{16}OH$, and 10 ms for other elements (e.g., Ca, Ti, Ni, Ce, Yb, Th), which were monitored to assess potential contamination by extraneous material. Data reduction was performed using the "Rb-Sr isotopes" data reduction scheme in Iolite 4 (Paton et al., 2011). Instrumental drift and quantification of $^{87}Sr/^{86}Sr$ and 'uncorrected' $^{87}Rb/^{86}Sr$ were undertaken by repeated ablation of NIST610, which was measured every block of 15 unknowns including in-house mica standards (see below). NIST610 is a synthetic glass with different ablation properties than mica and, therefore, this approach provides biased (i.e. 'uncorrected') $^{87}Rb/^{86}Sr$ ratios in mica analyses (e.g., Redaa et al., 2021). Correction of NIST610-based 'uncorrected' $^{87}Rb/^{86}Sr$ in the mica unknowns was performed following the method outlined by Giuliani et al. (2024). The calculated age of an in-house mica standard from the Wimbledon kimberlite (South Africa), which has a robustly constrained Rb-Sr age of 114.5 ± 0.8 Ma (2σ s.d.) based on isotope dilution analyses (Sarkar et al., 2023) and exhibits large variation in Rb/Sr (almost 3 orders of magnitude), was employed to calculate a correction factor that is then employed to obtain the final $^{87}Rb/^{86}Sr$ in the mica unknowns. The validity of this approach was confirmed by analyses of micas from the Bultfontein kimberlite (South Africa) and Mount Dromedary monzonite (MD-2; Australia) which returned Rb-Sr ages that are indistinguishable from solution-mode Rb-Sr and Ar-Ar analyses of mica on the same sample: 88.3 ± 0.2 Ma (Fitzpayne et al., 2020), and 99.20 ± 0.08 Ma (Phillips et al., 2017), respectively (**Supplementary Table S2**). Time-resolved spectra of mica unknowns and reference materials were screened to remove anomalous regions based on e.g., low concentrations of Rb and high concentrations of Sr, Ca, Ce and/or other incompatible trace elements. Analyses with total signals of less than 10 seconds (after screening) and with anomalously low contents of Rb or high contents of Sr (and Ca), often resulting in $^{87}Rb/^{86}Sr$ <2.5, as well as analyses with large $^{87}Sr/^{86}Sr$ uncertainties and data points that plotted distinctly off the isochron were not included in the Nicolaysen diagrams (**Supplementary Table S3**). Trace element concentrations were not quantified.



# Results

## Epidote Sr isotopes

$^{87}Sr/^{86}Sr$ ratios were measured by laser ablation MC-ICP-MS in 5 of the 10 samples employed for mica Rb-Sr geochronology. Two additional samples (SYGR44; SYGR50) were also included to corroborate the signature of the blueschist and metagabbro rocks. For comparison, we also present isotope-dilution Sr isotope data in samples from the Megas Gialos locality, including the 3 samples analyzed for epidote and mica Rb-Sr isotopes. A summary of the new and available Sr isotope data for epidote is reported in **Figure 3** and the full datasets, including bulk rock Sr and Nd isotopic compositions, are included in **Supplementary Tables S1 and S4.**

At Grizzas (Kampos belt), the two blueschist samples (SYGR36 and 44) show very small ranges in epidote $^{87}Sr/^{86}Sr$ compositions **(see Supplementary Figure S2)** with indistinguishable weighted means of 0.70805 ± 0.00006 (2SE; n = 12) and 0.70802 ± 0.00005 (2SE; n = 18; **Table 2 and Supplementary Figure S2**). The other two Grizzas samples (the metasomatic rind SYGR38 and the metagabbro SYGR50) exhibit larger isotopic variations. $^{87}Sr/^{86}Sr$ in sample SYGR38 vary widely between 0.70426 ± 0.00008 and 0.710002 ± 0.00008 (n = 22) with no statistically distinct populations (**Figure 3**). The weighted mean (although statistically meaningless) is similar to those of SYGR36 and SYGR44: 0.70767 ± 0.00058. In sample SYGR50, 16 epidote grains parallel to the foliation yield a restricted range in Sr isotope values corresponding to a weighted mean of 0.70460 ± 0.00004, which is substantially less radiogenic than the blueschist samples from Grizzas, although similar to the lowest $^{87}Sr/^{86}Sr$ of sample SYGR38. Four epidote grains within boudin necks of sample SYGR50 show more radiogenic values of up to 0.70585 ± 0.00020.

Epidote in the three samples from Megas Gialos show very limited within-sample $^{87}Sr/^{86}Sr$ variability with weighted means of 0.70466 ± 0.00004 (n = 24) for SYMG02; 0.70534 ± 0.00005 (n = 25) for SYMG07 and 0.70520 ± 0.00005 (n = 31) for SYMG08.03 The epidote Sr isotope compositions are not correlated with the lithology as the greenschist sample SYMG07 has the same $^{87}Sr/^{86}Sr$ as one of the two dilational veins (SYMG02 and 08.03). Measured (i.e. present-day) $^{87}Sr/^{86}Sr$ of bulk rock SYMG07 is 0.705414 ± 0.000008 (2σ s.d. of NBS987 standards measured in the same session), marginally more radiogenic than the SYMG07 epidote, and minimally affected by radiogenic ingrowth (e.g., ~0.0002 in 50 Myr) due to low bulk-rock $^{87}Rb/^{86}Sr$ of 0.290 (**Supplementary Table S1**). The bulk-rock $^{87}Sr/^{86}Sr$ of SYMG08.03 (0.705281 ± 0.000006) is almost indistinguishable from the epidote value reported above. The very low $^{87}Rb/^{86}Sr$ (0.073) suggests minimal radiogenic Sr ingrowth in this bulk sample.



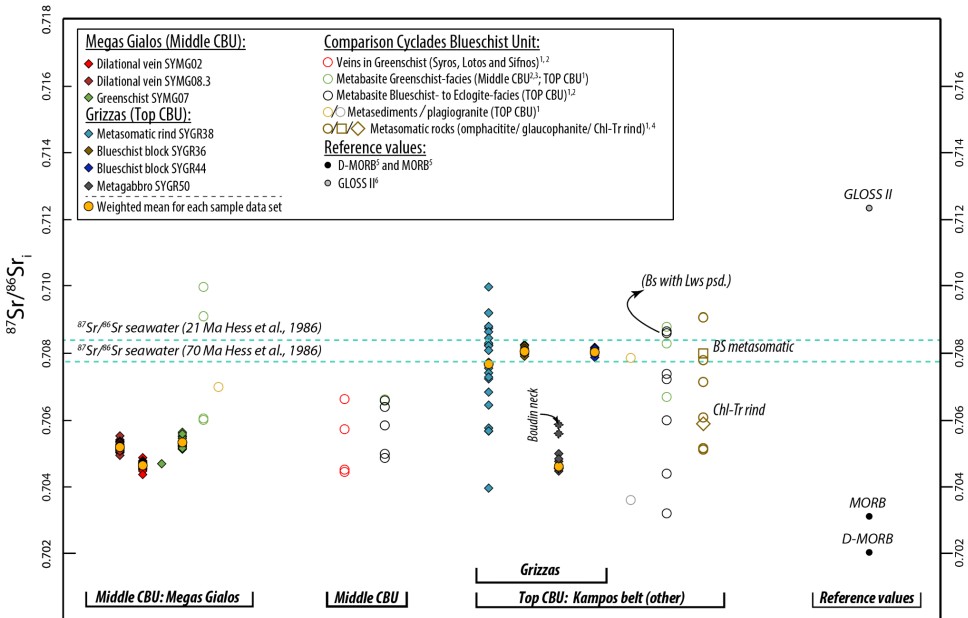

453

**Figure 3**. *Overview of $^{87}Sr/^{86}Sr$ in-situ laser ablation MC-ICP-MS epidote data points and comparison to ID-TIMS (whole rock and multi-mineral) analyses from different localities in Syros. The resulting $^{87}Sr/^{86}Sr$ values are assumed to represent initial ratios due to the lack of Rb in epidote. For comparison, pristine MORB and D-MORB, as well as compiled trench filling sediments (GLOSS II) along with Cretaceous to Miocene $^{87}Sr/^{86}Sr$ seawater values are shown. 1 – Glodny and Ring (2022); 2 – Kotowski et al. (2022); 3 – Bröcker et al. (2013); 4 – Bröcker and Enders (2001); 5 – Salters and Stracke (2004); 6 – Plank (2014).*

## Mica Rb-Sr dating

In this section we report the mica Rb-Sr isotope data and describe the related isochronous array for each sample, complemented in 5 cases by epidote Sr isotope results. The complete white mica dataset, including Rb and Sr isotope ratios, is provided in **Supplementary Table S3** (see **Table 2 for** a summary of the age data). For each sample we also provide a model age where the mica Rb/Sr isochron is anchored to an assumed $^{87}Sr/^{86}Sr$ value, that is $0.7080 \pm 0.0005$ for all the samples from Grizzas, and $0.7050 \pm 0.0005$ for those from Megas Gialos. For Grizzas, employing this value is justified by the fact that the weighted mean of epidote Sr isotopes are ~0.708 for three or the four analysed samples (**Figure 3;** see also the compiled data in **Figure 3** for metabasites from the Top CBU), and "unanchored" mica Rb/Sr isochrons are generally within uncertainty of this value (see below). The epidote and bulk-rock compositions at Megas Gialos cluster at $^{87}Sr/^{86}Sr$ of ~0.705 (**Figure 3**) hence providing a robustly constrained initial Sr composition for anchoring the mica-based Nicolaysen arrays. In the discussion section we will address the impact of changing initial (or "common") $^{87}Sr/^{86}Sr$ composition in the calculated Rb/Sr ages.





## SYGR36

White mica in the blueschist block sample SYGR36 show a spread in $^{87}Rb/^{86}Sr$ between 15 and 53 (n = 43/45) associated with variations in $^{87}Sr/^{86}Sr$ between 0.7166 and 0.7407 (**Figure 4A**). The limited Rb/Sr spread results in a poorly defined "unanchored" isochron age of 29 ± 17 Ma (2se, MSWD = 0.89, initial $^{87}Sr/^{86}Sr$ = 0.7149 ± 0.0062). Anchoring the phengite Rb-Sr data to epidote from the same sample (weighted mean $^{87}Sr/^{86}Sr$ = 0.70805 ± 0.00006) provides a rather different (although within uncertainty) and considerably more precise age of 47.2 ± 4.4 Ma (2s, MSWD = 4.2). Assuming a modeled initial $^{87}Sr/^{86}Sr$ of 0.7080 ± 0.0005 (1s) provides an age of 46.9 ± 5.1 Ma (2s, MSWD = 0.96), overlapping closely with the epidote-anchored isochron age.

## SYGR37

White mica grains in the metasomatic rind sample SYGR37 show a slightly larger spread in $^{87}Rb/^{86}Sr$ (22-112, n = 36/38) and $^{87}Sr/^{86}Sr$ (0.7211-0.7676) compared to SYGR36, resulting in a more precise unanchored isochron age of 32.3 ± 7.5 Ma (MSWD = 0.51, initial $^{87}Sr/^{86}Sr$ = 0.7158 ± 0.0059; **Supplementary Figure S3**). Anchoring these mica Rb-Sr to modeled initial $^{87}Sr/^{86}Sr$ of 0.7080 ± 0.0005 yields an older age of 41.1 ± 3.1 Ma (MSWD = 0.66).

## SYGR38

White mica in the metasomatic rind SYGR38 shows spreads between 22-90 and 0.7140-0.7771 for $^{87}Rb/^{86}Sr$ and $^{87}Sr/^{86}Sr$, respectively (n = 20/25, with 5 analyses excluded based on short signals of less than 10 seconds). The corresponding unanchored isochron age is 43 ± 10 Ma (MSWD = 0.6, initial $^{87}Sr/^{86}Sr$ = 0.7075 ± 0.0064; **Figure 4B**). Adding epidote Sr isotopes (weighted mean $^{87}Sr/^{86}Sr$ = 0.70767 ± 0.00058) to the mica Rb-Sr isochron yields the same, yet more precise age of 43.0 ± 5.4 Ma (MSWD = 480). Using a modeled initial $^{87}Sr/^{86}Sr$ of 0.7080 ± 0.0005 (1s) results in a similar age of 42.5 ± 5.5 Ma (MSWD = 0.54). Considering the large spread in epidote $^{87}Sr/^{86}Sr$ values (~0.7043 to ~0.7100), we have also calculated model ages using initial $^{87}Sr/^{86}Sr$ of 0.7050 and 0.7100 and these are within uncertainty of each other: 46.5 ± 5.6 Ma (MSWD = 0.57) and 39.8 ± 5.5 Ma (MSWD = 0.57), respectively( **Supplementary Figure S4)**.

GEOCHRONOLOGY
Discussions

EGU



***Figure 4****. Representative laser-ablation ICP-MS/MS Rb-Sr isochrons of white micas from Grizzas (North East*
*Syros Island, SYGR, A-D) and Megas Gialos (South Syros Island, SYMG, E-F). The size of the ellipses represents*
*internal 2 SE (standard error), where data points that were excluded from the regression are displayed as empty*
*ellipses. Isochronous regressions are plotted together with their 95% confidence level envelopes in different*



*colours based on the employed anchoring technique: purple for mica-only unanchored regressions; blue for*
*regressions anchored to epidote; orange for regressions anchored to a modelled initial $^{87}Sr/^{86}Sr$ of 0.7080 ±*
*0.0005. The number below the sample labels indicates the number of mica analyses. All plots were generated*
*using IsoplotR (Vermeesch, 2018).*

### SYGR41

White mica from the dilational vein SYGR41 show a limited spread in $^{87}Rb/^{86}Sr$ between 14-
63 (n = 36/36) associated with variations in $^{87}Sr/^{86}Sr$ between 0.7116 and 0.7498. An
unanchored isochron through these data yields an age of 45 ± 11 Ma (MSWD = 0.78, initial
$^{87}Sr/^{86}Sr = 0.7076 ± 0.0049$). Anchoring these mica Rb-Sr data to a modeled initial $^{87}Sr/^{86}Sr$ of
0.7080 ± 0.0005 (1s) yields the same, yet more precise age of 44.7 ± 4.5 Ma (2s, MSWD =
519   0.74).


### SYGR42

$^{87}Rb/^{86}Sr$ and $^{87}Sr/^{86}Sr$ ratios in phengites from the metasomatized metagabbro sample
SYGR42 range from 27 to 185 and 0.7155 to 0.8162, respectively (n = 30/30), and the
corresponding unanchored isochrons provides an age of 46 ± 9 Ma (MSWD = 1.3, initial
$^{87}Sr/^{86}Sr = 0.7090 ± 0.0079$). Anchoring these mica Rb-Sr data to $^{87}Sr/^{86}Sr = 0.7080 ± 0.0005$
(1s) results in an overlapping, although more precise age of 46.6 ± 4.6 Ma (MSWD = 1.2).

### SYGR45

Two generations of phengite laths, parallel and oblique to the main foliation, from the
metasedimentary rock sample SYGR45 display $^{87}Rb/^{86}Sr$ values between 16 and 90 (n = 62/65)
and a corresponding variation in $^{87}Sr/^{86}Sr$ between 0.7134 and 0.7647, with no systematic
difference between the two textural types of mica (**Figure 4C**; **Supplementary Table S3**). The
resulting unanchored isochron has a slope equivalent to an age of 44.8 ± 8.6 Ma (MSWD = 1,
initial $^{87}Sr/^{86}Sr = 0.7092 ± 0.0038$). Anchoring these mica Rb/Sr data to a modeled initial
$^{87}Sr/^{86}Sr$ of 0.7080 ± 0.0005 (1s) yields a slightly older and more precise age of 47.2 ± 3.8 Ma
(MSWD = 1), overlapping with the unanchored age within uncertainty. Using a more
radiogenic initial $^{87}Sr/^{86}Sr$ of 0.7100 has a small effect on the calculated age (43.2 ± 3.8;
MSWD = 1).

### SYGR58

The two textural types of white mica identified in the felsic pod sample SYGR58, parallel and
oblique to the main foliation, exhibit indistinguishable Rb-Sr isotope systematics
(**Supplementary Table S3**) and are, hence, described together. These white micas show the
largest Rb/Sr spread observed in the sample set of between 8 and 671 (n = 22/58, where only
analyzes with $^{87}Rb/^{86}Sr > 2.5$ were considered for the isochron), which is consistent with the
felsic nature of this sample. The spread in $^{87}Sr/^{86}Sr$ is between 0.670 and 1.073, resulting in
precise, although unanchored Rb/Sr age of 36.9 ± 2.4 Ma (MSWD = 0.67, initial $^{87}Sr/^{86}Sr =$





0.7038 ± 0.0072; **Figure 4D**). Anchoring these mica Rb/Sr data to a modeled initial $^{87}$Sr/$^{86}$Sr
of 0.7080 ± 0.0005 (1s) yields a similar age of 36.1 ± 2.1 Ma (2s, MSWD = 0.67).

## SYMG02

Phengites from the dilational vein sample SYMG02 show a relatively large $^{87}$Rb/$^{86}$Sr spread
between 14 and 195 (n = 23/25) associated with a restricted $^{87}$Sr/$^{86}$Sr spread between 0.6976
and 0.7944. These data define a meaningless unanchored isochron (age = 11 ± 11 Ma, MSWD
= 1, initial $^{87}$Sr/$^{86}$Sr = 0.723 ± 0.021; **Figure 4E**). Adding epidote Sr data from the same sample
(weighted mean $^{87}$Sr/$^{86}$Sr = 0.70466 ± 0.00004) to the mica Rb-Sr isotopes results in a more
meaningful age of 20.0 ± 5.1 Ma (MSWD = 3.2). Using a modeled initial $^{87}$Sr/$^{86}$Sr anchor of
0.7050 ± 0.0005 (1s) yields a similar age of 19.8 ± 5.2 Ma (2s, MSWD = 1).

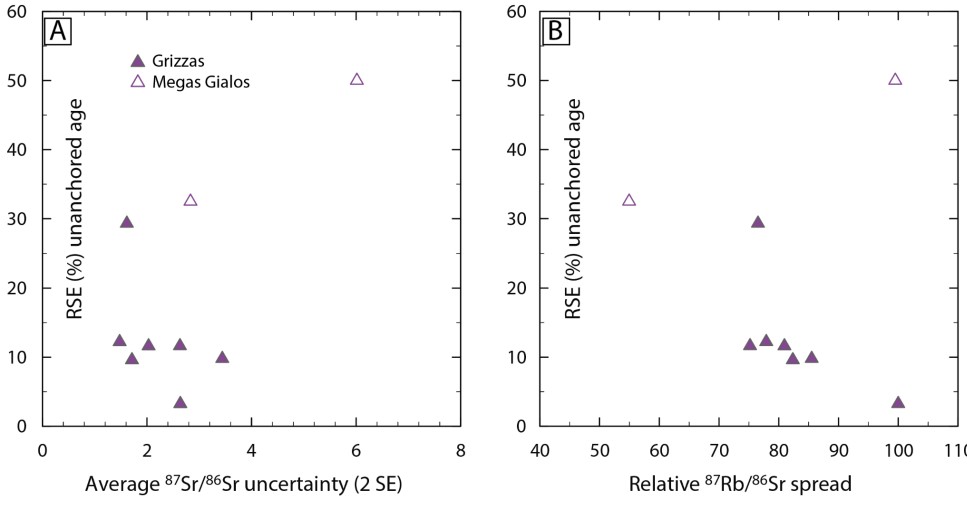

**Figure 5.** *Comparison of relative standard deviations (1 SE, standard error) of unanchored mica Rb/Sr ages and (A) average $^{87}$Sr/$^{86}$Sr uncertainties and (B) relative (%) $^{87}$Rb/$^{86}$Sr spread. The latter was defined as the ratio between the absolute $^{87}$Rb/$^{86}$Sr spread and the highest $^{87}$Rb/$^{86}$Sr value observed for any given sample, resulting in a number between 0 and 100%.*

## SYMG07

$^{87}$Rb/$^{86}$Sr and $^{87}$Sr/$^{86}$Sr values in white mica from the greenschist sample SYMG07 vary
between 75-231 and 0.7199-0.8121, respectively (n = 12/13), yielding a meaningless isochron
age of 6.1 ± 31.2 Ma (MSWD = 1.2, initial $^{87}$Sr/$^{86}$Sr = 0.749 ± 0.063; **Supplementary Figure**
**S3**). Coupling white mica with the SYMG07 epidote data (weighted mean $^{87}$Sr/$^{86}$Sr = 0.70534
± 0.00005) results in an age of 27.1 ± 8.4 Ma (MSWD = 3.4). An identical age is obtained
using a model initial $^{87}$Sr/$^{86}$Sr of 0.7050 ± 0.0005 (1s): 27.4 ± 8.4 Ma (2s, MSWD = 1.2).





**572**  **SYMG08**

**573**  Phengites from dilational vein SYMG08.3 show a spread in $^{87}$Rb/$^{86}$Sr between 45 and 123 (n
**574**  = 30/30) associated with variations in $^{87}$Sr/$^{86}$Sr between 0.7084 and 0.7606 (**Figure 4F**). These
**575**  data define an unanchored isochron age of 20 ± 13 Ma (MSWD = 0.8, initial $^{87}$Sr/$^{86}$Sr = 0.706
**576**  ± 0.015). Adding Sr epidote data (weighted mean $^{87}$Sr/$^{86}$Sr = 0.70520 ± 0.00005) to the
**577**  phengite Rb/Sr data results in the same, yet more precise age of 20.6 ± 3.0 Ma (MSWD = 2.7).
**578**  The results hardly change by anchoring the mica Rb/Sr data to a modeled initial $^{87}$Sr/$^{86}$Sr of
**579**  0.7050 ± 0.0005 (1s): 20.8 ± 3.1 Ma (MSWD = 0.75) (**Supplementary Figure S4**).

Table 2. Summary of epidote Sr isotopes and mica Rb-Sr ages

| Sample ID | Epidote $^{87}$Sr/$^{86}$Sr * | | | Mica analyses | Mica age, unanchored ** | | | | Mica + epidote age (Ma) | | | Mica age (Ma), anchored *** | | |
|---|---|---|---|---|---|---|---|---|---|---|---|---|---|---|
| | n | mean | 2 SE | n | age (Ma) | 2 SE | MSWD | isochron y intercept | age (Ma) | 2 SE | MSWD | age (Ma) | 2 SE | MSWD |
| *Grizzas (NE Syros)* | | | | | | | | | | | | | | |
| SYGR36 | 12 | 0.70805 | 0.00006 | 43/45 | 29 | 17 | 0.89 | 0.7149 ± 0.0062 | 47.2 | 4.4 | 4.2 | 46.9 | 5.1 | 0.96 |
| SYGR37 | – | – | – | 36/38 | 32.3 | 7.5 | 0.51 | 0.7158 ± 0.0059 | – | – | – | 41.1 | 3.1 | 0.66 |
| SYGR38 | 21 | 0.70767 | 0.00058 | 20/25 | 43 | 10 | 0.60 | 0.7075 ± 0.0064 | 43.0 | 5.4 | 480 | 42.5 | 5.5 | 0.54 |
| SYGR41 | – | – | – | 36/36 | 45 | 11 | 0.78 | 0.7076 ± 0.0049 | – | – | – | 44.7 | 4.5 | 0.74 |
| SYGR42 | – | – | – | 30/30 | 46 | 9 | 1.30 | 0.7090 ± 0.0079 | – | – | – | 46.6 | 4.6 | 1.2 |
| SYGR44 | 18 | 0.70802 | 0.00005 | – | – | – | – | – | – | – | – | – | – | – |
| SYGR45 | – | – | – | 62/65 | 44.8 | 8.6 | 1.00 | 0.7092 ± 0.0038 | – | – | – | 47.2 | 3.8 | 1 |
| SYGR50 | 16 | 0.70460 | 0.00004 | – | – | – | – | – | – | – | – | – | – | – |
| SYGR58 | – | – | – | 22/58 | 36.9 | 2.4 | 0.67 | 0.7038 ± 0.0072 | – | – | – | 36.1 | 2.1 | 0.67 |
| *Megas Gialos (SE Syros)* | | | | | | | | | | | | | | |
| SYMG02 | 24 | 0.70466 | 0.00004 | 23/25 | 11 | 11 | 1.00 | 0.723 ± 0.021 | 20.0 | 5.1 | 3.2 | 19.8 | 5.2 | 1 |
| SYMG07 | 25 | 0.70534 | 0.00005 | 12/13 | 6.4 | 31.2 | 1.20 | 0.749 ± 0.063 | 27.1 | 8.4 | 3.4 | 27.4 | 8.4 | 1.2 |
| SYMG08.3 | 31 | 0.70520 | 0.00005 | 30/30 | 20 | 13 | 0.80 | 0.706 ± 0.015 | 20.6 | 3.0 | 2.7 | 20.8 | 3.1 | 0.75 |

\* laser ablation, multi-collector ICP-MS; complete dataset in Supplementary Table S1
\*\* laser ablation, ICP-MS/MS; complete dataset in Supplementary Table S3
**580**  \*\*\* anchoring values: 0.7080 ± 0.0005 for SYGR samples; 0.7050 ± 0.0005 for SYMG samples

**581**  # Discussion

**582**  ## Optimal strategies to obtain robust Rb-Sr ages for white
**583**  ## mica in young metamorphic rocks by LA-ICP-MS/MS

**584**  White mica in all the investigated samples, and regardless of their bulk-rock chemistry (i.e.
**585**  mafic and metasomatic), exhibit limited spread in Rb/Sr compared to previous studies (e.g.,
**586**  Kirkland et al., 2023, Glodny and Ring 2022). Except for the relatively large spread observed
**587**  in the felsic sample SYGR58 ($^{87}$Rb/$^{86}$Sr = 8 to 671), the Rb/Sr range of all the other samples
**588**  never exceeds one order of magnitude and in some cases less (e.g., $^{87}$Rb/$^{86}$Sr = 15-53 in
**589**  blueschist SYGR36) compared to, for example, the two to three orders of magnitude in
**590**  phlogopite from lamproites and kimberlites (Giuliani et al., 2024), or biotite in some
**591**  metamorphosed granites (Ceccato et al., 2024). In addition, the combination of relatively low
**592**  Rb contents (not quantified but inferred from low Rb/Sr ratios) and geologically young
**593**  (Cenozoic) age of the Syros micas did not allow the ingrowth of substantial radiogenic $^{87}$Sr as
**594**  shown by the low measured $^{87}$Sr/$^{86}$Sr (generally <0.8; **Supplementary Table S3**). Low $^{87}$Sr
**595**  contents are associated with large uncertainties for $^{87}$Sr/$^{86}$Sr, which systematically exceed 1%
**596**  (2SE) for individual measurements (**Figure 5A**). The compounded effects of low absolute
**597**  $^{87}$Rb/$^{86}$Sr values (generally <200 and, for some samples, <100), limited spread in Rb/Sr and
**598**  poor precision in the quantification of $^{87}$Sr/$^{86}$Sr result in large uncertainties associated with the
**599**  slopes of unanchored mica Rb-Sr isochrons. These uncertainties translate to a poor precision




for the related ages with 10-29 %RSE (relative standard error) in the SYGR samples (except
for the felsic sample SYGR58, with an RSE of 3%, i.e. 36.9 ± 2.4 Ma, 2SE), and even larger
for the younger SYMG samples (**Figure 6A**). The inverse correlation between relative
$^{87}Rb/^{86}Sr$ spread and age uncertainty of unanchored isochrons in **Figure 5B** exemplifies the
impact of Rb/Sr variations on age precision. In at least three cases (SYGR36, SYMG02 and
SYMG07) these unanchored mica-only isochronous arrays are not just imprecise, but also
rather inaccurate as shown by the substantially older ages of the mica + epidote isochron for
SYGR36 (29 ± 17 Ma vs 47.2 ± 4.4 Ma for SYGR36) or simply geologically meaningless (11
± 11 Ma and 6.4 ± 31 Ma for SYMG02 and SYMG07, respectively; **Table 2**).
To overcome the limitations in mica Rb-Sr geochronology by LA-ICP-MS/MS due to low
Rb/Sr and/or young ages, the two viable solutions explored here include anchoring the
isochronous arrays to either the Sr isotope composition of a low Rb/Sr phase in textural (and
probably chemical equilibrium) with mica, such as epidote, or an assumed $^{87}Sr/^{86}Sr$ value. The
latter approach effectively provides a "model age" and, while previously explored by Rösel
and Zack (2021), it is rigorously evaluated herein by a systematic comparison with initial Sr
isotope constraints from epidote and bulk rocks. Anchoring mica isochrons to a low Rb/Sr
phase has been rarely applied in mica Rb/Sr geochronology by LA-ICP-MS/MS (Barnes et al.,
2024; Giuliani et al., 2024), while being widely employed for conventional Rb/Sr dating by
isotope dilution (e.g., Maas, 2003; Glodny et al., 2008; Hyppolito et al., 2016; Angiboust et al.,
2018; Dalton et al., 2020). Comparisons of unanchored mica Rb/Sr ages with those anchored
using mean $^{87}Sr/^{86}Sr$ of epidote analyses show an improvement in precision of up to 6 times
(**Figure 6B**) – as well as better accuracy in some cases as shown above for SYGR36. Clearly,
in young high-pressure metamafic rocks such as those from Syros, this approach is
recommended to obtain robust age constraints even when the limited spread in mica Rb/Sr
prevents generation of meaningful isochronous arrays (i.e. SYMG02 and 07).

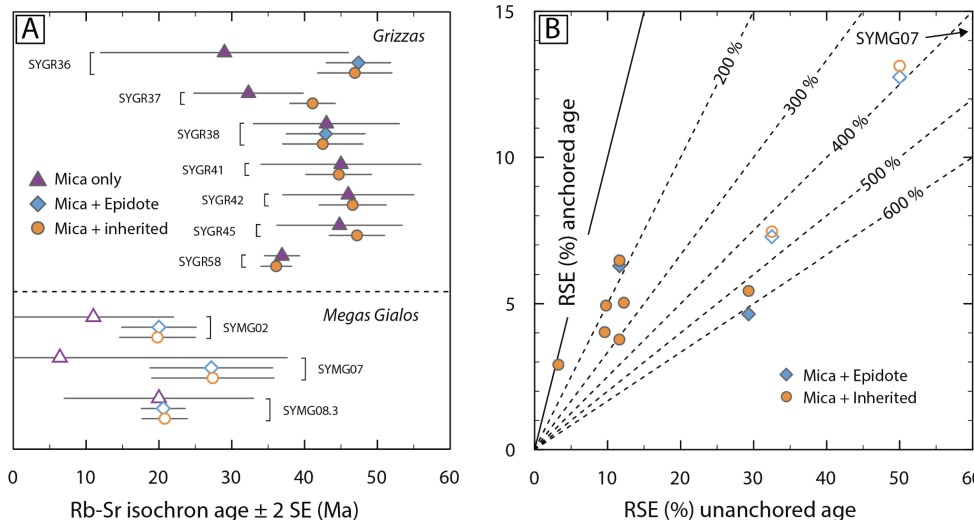

**Figure 6.** (A) Overview of mica Rb-Sr ages using mica datapoints only (unanchored isochrons, purple), anchoring
to epidote (blue) and anchoring to an assumed initial $^{87}Sr/^{86}Sr$ composition (orange). Initial $^{87}Sr/^{86}Sr$ was assumed





*to be 0.7080 ± 0.0005 for Grizzas and 0.7050 ± 0.0005 for Megas Gialos (see text). (B) Comparison of the*
*uncertainties expressed as % RSE (relative standard error) for unanchored mica-only ages and ages anchored to*
*either epidote or an assumed initial ⁸⁷Sr/⁸⁶Sr. Samples from Grizzas and Megas Gialos are shown as empty and*
*full symbols, respectively. Location of Megas Gialos sample SYMG07 (unanchored age 6.4 ± 31 Ma, 243% RSE)*
*is shown with an arrow..*
Model ages are also, not surprisingly, substantially more precise than unanchored mica-only
Rb/Sr ages. However, their accuracy deserves scrutiny. Where epidote data are available, the
model ages calculated in this work can be employed to show the effect of inaccurate initial
$^{87}$Sr/$^{86}$Sr in the isochron ages (**Figure 7**). At Grizzas, epidote $^{87}$Sr/$^{86}$Sr varies between 0.7043
and 0.7100 (all this variation is contained in the metasomatic rind sample SYGR38). Using
available bulk rock data for the Kampos Belt (**Figure 3**), this range can be extended downward
to ~0.7030, hence effectively bracketing the possible compositions of initial Sr to calculate
mica model ages. For simplicity, the same range is employed for Megas Gialos. Beyond the
model ages presented in the results section and **Table 2**, for each sample two additional model
ages are calculated using an initial $^{87}$Sr/$^{86}$Sr of 0.7030 ± 0.005 and 0.7100 ± 0.005, respectively
(**Figure 7 and Supplementary Table S5**). In the Grizzas samples, using an initial $^{87}$Sr/$^{86}$Sr of
0.7100 generates model ages that are generally within uncertainty of those where the initial
$^{87}$Sr/$^{86}$Sr was assumed to be 0.7080; conversely, the ages are ≥10% older
if an initial $^{87}$Sr/$^{86}$Sr value of 0.7030 is employed. **Figure 7** shows that the older the
sample, the more dramatic is the impact of the initial $^{87}$Sr/$^{86}$Sr chosen. For the > 40 Ma Grizzas
micas, the use of initial $^{87}$Sr/$^{86}$Sr of 0.7100 provides ages that are resolvable (i.e. outside 2SE)
from those obtained employing 0.7030 as the initial $^{87}$Sr/$^{86}$Sr ratio. Conversely, for the younger
(<30 Ma) Megas Gialos samples, all the calculated model ages are within uncertainty of each
other. While the favored approach remains to analyze a low Rb/Sr phase cogenetic to mica
(e.g., epidote, plagioclase, carbonate, apatite), where there is limited independent knowledge
of initial Sr isotope compositions, we recommend employing $^{87}$Sr/$^{86}$Sr that are intermediate
between those of likely endmembers representative of the examined lithologies.

At Grizzas, the blueschist blocks (samples SYGR36 and SYGR44) and a metasomatic rind
(sample SYGR38) consistently yielded initial $^{87}$Sr/$^{86}$Sr values close to 0.708, although the latter
shows scattering between 0.704 to 0.710 (**Figure 3**). In the literature, highly radiogenic values
in metamafic and metasomatic rocks are common in the Kampos Belt, including for some
metasedimentary rocks (**Figure 3**). On the other hand, a metagabbro (sample SYGR50) yielded
an initial $^{87}$Sr/$^{86}$Sr value close to 0.705 Similarly, the metamafic greenschist (SYMG07) and
veins (SYMG02 and SYMG08.3), along with additional vein and greenschist samples analyzed
for bulk rock $^{87}$Sr/$^{86}$Sr only from Megas Gialos consistently yielded in-situ epidote and age-
corrected TIMS whole rock $^{87}$Sr/$^{86}$Sr values of c. 0.705 (**Figure 3 and supplementary Table
S4**). We interpret the least radiogenic values to represent the oceanic magmatic protolith (e.g.,
Taylor and Lasaga, 1999) as well as veins that have equilibrated with or sourced from
metamafic rocks. In contrast, the more radiogenic signature could have been introduced by pre-
subduction seafloor alteration (Voigt et al., 2021), or by metasomatism by highly radiogenic
fluids for example derived from dehydration of metasedimentary rocks (Halama et al., 2011).
The latter hypothesis is more consistent with the spatial association between metasedimentary





and metasomatic rocks within the Grizzas shear zone. Our results demonstrate that for high-
pressure metamafic rocks in subduction zones, the commonly assumed MORB-like $^{87}Sr/^{86}Sr$
value of 0.703 (Rösell and Zack, 2021) might not necessarily be representative of the initial Sr
isotope composition.

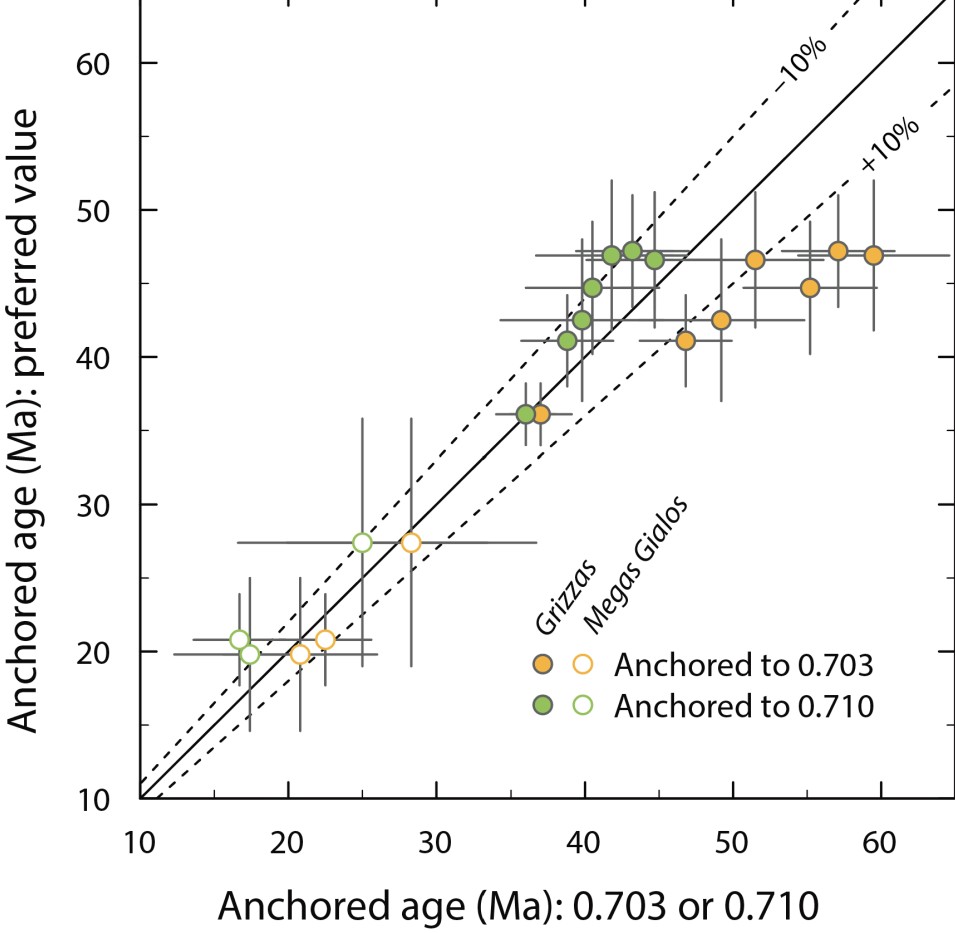


**Figure 7.** *Covariation plots showing the effect of assumed initial $^{87}Sr/^{86}Sr$ on the mica Rb-Sr "model" age.*
*Preferred anchoring values are 0.7080 ± 0.0005 for Grizzas and 0.7050 ± 0.0005 for Megas Gialos (vertical*
*axis), which are compared to the extreme values in the range of observed bulk rock data for the Kampos Belt:*
*0.7030 ± 0.0005 (orange) and 0.7100 ± 0.0005 (green) (horizontal axis).*

## Application to Syros

To further validate our newly acquired mica Rb/Sr ages (anchored to epidote or, when not
available, to a modeled initial $^{87}Sr/^{86}Sr$; **Table 2**), we compare them with published age
constraints from Kampos Belt (Top CBU) and Middle CBU localities (**Figure 8**). Kotowski et
al. (2022) and Glodny and Ring (2022) compiled and reported new ID TIMS Rb-Sr ages,



mostly from the Western Kampos Belt and outcrops along the Top CBU in Syros, ranging from
53 to 43 Ma. This age range is interpreted to date the eclogite-to-blueschist-facies subduction
fabrics, developed during the prograde-to-peak-pressure and earliest stage of exhumation.
Robust U-Pb zircon and Lu-Hf garnet ages between 53 and 48 Ma constrain the peak
metamorphism in the Grizzas area (see Tomascheck et al., 2003; Lagos et al., 2007), and are
in agreement with the higher end of the Rb-Sr multi-mineral isochron ages including white
mica separates (e.g., Glodny and Ring 2022). Recent in-situ Rb-Sr dating of white mica also
showed an age of 48.4 ± 3.6 Ma for an eclogite from the Kathergaki cape (presumably
belonging to the Top CBU), which was also interpreted to date the near-peak metamorphism
(Barnes et al., 2024). At Grizzas, a blueschist block (SYGR36), a metasomatized metagabbro
(SYGR42) and a metasediment (SYGR45) yielded mica Rb/Sr ages ranging from 46.6 ± 4.6
Ma to 47.2 ± 3.8 Ma (**Table 2** and **Figure 6**). Similarly, the dilatational vein sample SYGR41
returned a mica Rb/Sr age consistent with the HP metamorphic stage (44.7 ± 4.5 Ma). These
ages overlap with the low-end of the HP eclogite-to-blueschist-facies near-peak metamorphism
(peak to the earliest exhumation). Thus, and in line with previous investigations, the obtained
ages are interpreted to date near-peak metamorphism (for the blueschist SYGR36 and
metasediment SYGR45 samples) as well as the oldest record of near-peak metasomatism and
shear zone development leading to veining (SYGR41) and metagabbro fluid-assisted
deformation (SYGR42).

Kotowski et al. (2022) and Glodny and Ring (2022) noted that ages for the retrograde stage
associated with early decompression in the epidote blueschist-facies are in the 45 to 40 Ma
range, which could also be related to a mixed signal due to partial re-equilibration between the
early lawsonite blueschist- and HP greenschist-facies metamorphism (Glodny and Ring, 2022).
Blueschist- to (HP)greenschist-facies retrogression during exhumation is constrained to occur
between 40 and 20 Ma in the Kampos Belt based on previous Rb-Sr and Ar-Ar geochronology
(Glodny and Ring, 2022; Kotowski et al., 2022; Laurent et al., 2017). The metasomatic rind
samples SYGR37 and SYGR38 yielded mica Rb/Sr ages more consistent with fluid
metasomatism during the early exhumation stage in the epidote blueschist-facies stability field
(41.1 ± 3.1 Ma and 43.0 ± 5.4 Ma), although sample SYGR38 could be similarly interpreted to
date the metasomatism at near-peak pressure conditions considering the age uncertainty. These
c. 43 and 41 Ma ages date continuous fluid-rock interaction during HP deformation, which
preferentially occurs along shear zones (Zack and John, 2007; Angiboust et al., 2014; Kleine
et al., 2014; Smit and Pogge von Strandmann, 2020; Rajič et al., 2024). Only one sample (felsic
pod SYGR58) shows a statistically younger age of 36.1 ± 2.1 Ma, which is within the period
of exhumation and transition from blueschist to HP greenschist-facies. This age is consistent
with petrographic evidence of chlorite pseudomorphs after garnet suggestive of selective
greenschist-facies retrogression.
Overall, our near-peak ages align with two ages (samples 9C and 27; see **Figure 8**) reported
by Gyomlai et al. (2023a) for metasomatic lithologies within the Kampos Belt (Lia side), while
our HP early exhumation ages are comparable, within uncertainty, to one of their ages (sample
9A) – however, the significantly larger uncertainties of their mica Rb/Sr ages for similar rock
types should be noted. Additionally, Gyomlai et al. (2023a) obtained three ages of c. 36 Ma
(samples Ln57, Ln10 and Ln1), overlapping with our sample SYGR58 (felsic pod), which they



interpreted as retrograde ages dating the "main" metasomatic event along Kampos. Our data
points to at least one event of HP metasomatism within the range of 47.2 ± 3.8 Ma to 41.1 ±
3.1 Ma, however, due to method uncertainties, distinguishing between multiple events within
this time range is not feasible. Furthermore, Barnes et al. (2024) reported an in-situ white mica
Rb-Sr age of 44.5 ± 3.1 Ma for a metasomatic eclogite (Delfini locality; presumably Middle
CUB). Thus, our data, along with the results from Barnes et al. (2024) are at odds with previous
interpretations which suggested that metasomatism along the entire Kampos Belt occurred as
a discrete pulse during the latest stages of exhumation (Gyomlai et al., 2023a). Instead, we
suggest that metasomatism along Kampos initiated at near-peak metamorphic conditions and
evolved through HP early exhumation. This enables us to constrain localized shear zone
activity under HP conditions within the subduction channel in the presence of fluids. These
metasomatic events may be temporally and spatially associated with processes such as deep
slicing, underplating, and slow slip and tremor (Angiboust et al., 2012; Behr et al. 2018; Agard
et al., 2018; Muñoz-Montecinos et al., 2020; Tewksbury-Christle et a., 2021; Behr and
Bürgmann, 2021).

In the Megas Gialos locality, the host greenschist sample yielded an age of 27.4 ± 8.4 Ma, in
line with previous investigations of lithologies from the Middle CBU which have shown ages
of greenschist-facies metamorphism younger than c. 35 Ma using Ar-Ar and ID TIMS Rb-Sr
geochronology (Glodny and Ring, 2022; Bröcker et al., 2013). The vein samples SYMG02 and
SYMG08.3 yielded potentially younger (although not statistically resolvable) ages of 19.8 ±
5.2 Ma and 20.8 ± 3.1 Ma, interpreted to date dilational veining during the latest stages of
exhumation of the metamorphic nappe at the base of the forearc (Cisneros et al., 2020; Muñoz-
Montecinos and Behr, 2023). These ages align with phengite + glaucophane veins from the
Top CBU unit (Elvia Island), which yielded virtually identical in-situ white mica Rb/Sr
(anchored to glaucophane) ages for dilational veining at conditions of c. 350 °C and 0.8 GP
(Ducharme et al., 2024). Thus, the finding of similar ages for transitional blueschist-to-
greenschist-facies dilational veining in Syros and in Evia Island demonstrates that across-dip
fluid flow toward the forearc is was ubiquitous process that occurred along the Hellenic
subduction zone at c. 20-22 Ma.



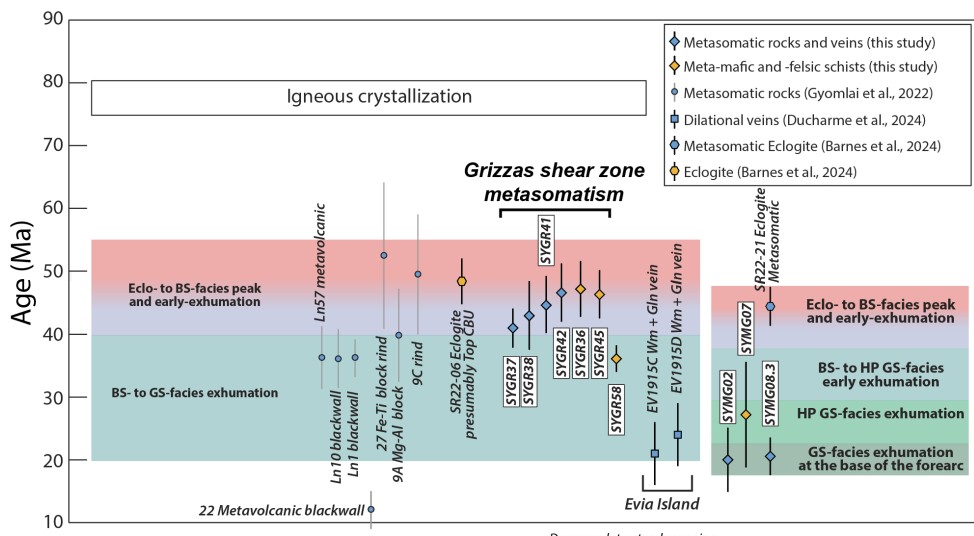

**Figure 8**. *Summary of in-situ mica Rb-Sr ages from this study along with previous investigations in Syros Island and other localities along the CBU (Evia Island). The fields depicting the timing of the main tectonometamorphic events represents a synthesis of the compilations from Kotowski et al. (2022) and Glodny and Ring (2022), to which the reader is referred for a more complete compilation of the geochronologic data collected in Syros and all along the CBU. BS – blueschist; Eclo – eclogite; GS – greenschist; HP – high pressure.*

# Summary

We systematically evaluated the limitations of mica Rb-Sr dating by LA-ICP-MS/MS for young metamafic samples using metamorphic rocks from Syros and attempted to circumvent these limitations by anchoring the initial $^{87}Sr/^{86}Sr$ component to either a low $^{87}Rb/^{86}Sr$ phase (i.e. epidote) or a modeled value. White mica analysis yielded narrow $^{87}Rb/^{86}Sr$ spread (ranging from 14 to 231 across the whole dataset), along with unradiogenic and imprecise $^{87}Sr/^{86}Sr$ (generally <0.8; 2SE typically exceeding 1%). The combined effect of low $^{87}Rb/^{86}Sr$ values, limited spread in Rb/Sr and high uncertainty in $^{87}Sr/^{86}Sr$ resulted in mica-only ages (i.e. without anchoring) with very large uncertainties of 10 to 35% RSE or higher in some cases.

By anchoring these data to a low Rb/Sr phase such as epidote, age precision improved by up to six times, aligning with previous Rb-Sr TIMS data from Syros and other localities along the Cyclades blueschists unit. A first set of samples yielded ages consistent with near-peak to early exhumation along the epidote-blueschist-facies. The youngest ages likely date the latest stage of (HP)greenschist-facies exhumation. These ages are interpreted as dating various metasomatic stages that likely initiated at near-peak metamorphic conditions and continued during exhumation. We noted unexpectedly high radiogenic $^{87}Sr/^{86}Sr$ values and sometimes variability for the metamafic-metasomatic materials. These values, likely resulting from focused fluid flow and metasomatism along the studied shear zone, underscore the importance


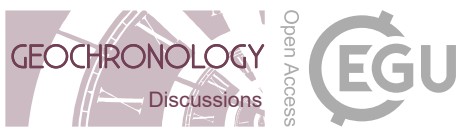

of carefully selecting and evaluating the geologic context of $^{87}$Sr/$^{86}$Sr anchors for future
applications of this "model" Rb-Sr white mica dating methodology.

# Data availability

All Laser Ablation ICP-MS/MS and MC-ICP-MS data is available in the supplementary
material.

# Author contribution

JM-M and AG designed the study and performed the experiments, with contributions from
BP. JM-M and WB collected the studied samples. AG and SO developed the statistical
analysis. JM-M and AG prepared the manuscript with contributions from all co-authors.

# Competing interests

The authors declare that they have no conflict of interest.

# Acknowledgments

We would like to thank Madalina Jaggi and Marcel Guillong for invaluable technical support
and Heather Stoll for granting access to the Agilent 8800 employed in this work. This project
was supported by the Swiss National Foundation (Ambizione fellowship n. PZ00P2_180126/1
to A. Giuliani) and ERC Starting Grant (947659) awarded to W.M. Behr.



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

investigation of hydrous cumulate melting in the southern Adamello batholith.
Contributions to Mineralogy and Petrology, 178(9), 64.
58. Plank, T. (2014). The chemical composition of subducting sediments. Elsevier.
59. Putlitz, B., Cosca, M. A., & Schumacher, J. C. (2005). Prograde mica 40Ar/39Ar
growth ages recorded in high pressure rocks (Syros, Cyclades, Greece). Chemical
Geology, 214(1-2), 79-98.
60. Redaa, A., Farkaš, J., Gilbert, S., Collins, A. S., Wade, B., Löhr, S., ... & Garbe-
Schönberg, D. (2021). Assessment of elemental fractionation and matrix effects
during in situ Rb–Sr dating of phlogopite by LA-ICP-MS/MS: implications for the
accuracy and precision of mineral ages. Journal of Analytical Atomic Spectrometry,
36(2), 322-344.
61. Ribeiro, B. V., Kirkland, C. L., Finch, M. A., Faleiros, F. M., Reddy, S. M., Rickard,
W. D., & Michael, I. H. (2023). Microstructures, geochemistry, and geochronology of
mica fish: Review and advances. Journal of Structural Geology, 104947.
62. Rubatto, D., Williams, M., Markmann, T. A., Hermann, J., & Lanari, P. (2023).
Tracing fluid infiltration into oceanic crust up to ultra-high-pressure conditions.
Contributions to Mineralogy and Petrology, 178(11), 79.
63. Salters, V. J., & Stracke, A. (2004). Composition of the depleted mantle.
Geochemistry, Geophysics, Geosystems, 5(5).
64. Sarkar, S., Giuliani, A., Dalton, H., Phillips, D., Ghosh, S., Misev, S., & Maas, R.
(2023). Derivation of Lamproites and Kimberlites from a Common Evolving Source
in the Convective Mantle: the Case for Southern African 'Transitional Kimberlites'.
Journal of Petrology, 64(7), egad043.
65. Schmidt, M. W., Vielzeuf, D., & Auzanneau, E. (2004). Melting and dissolution of
subducting crust at high pressures: the key role of white mica. Earth and Planetary
Science Letters, 228(1-2), 65-84.
66. Seman, S., Stockli, D. F., & Soukis, K. (2017). The provenance and internal structure
of the Cycladic Blueschist Unit revealed by detrital zircon geochronology, Western
Cyclades, Greece. Tectonics, 36(7), 1407-1429.
67. Smit, M. A., & von Strandmann, P. A. P. (2020). Deep fluid release in warm
subduction zones from a breached slab seal. Earth and Planetary Science Letters, 534,
116046.
68. Soukis, K., & Stockli, D. F. (2013). Structural and thermochronometric evidence for
multi-stage exhumation of southern Syros, Cycladic islands, Greece. Tectonophysics,
595, 148-164.
69. Tewksbury-Christle, C. M., Behr, W. M., & Helper, M. A. (2021). Tracking deep
sediment underplating in a fossil subduction margin: Implications for interface
rheology and mass and volatile recycling. Geochemistry, Geophysics, Geosystems,
22(3), e2020GC009463.
70. Tillberg, M., Drake, H., Zack, T., Kooijman, E., Whitehouse, M. J., & Åström, M. E.
(2020). In situ Rb-Sr dating of slickenfibres in deep crystalline basement faults.
Scientific reports, 10(1), 562.
71. Tillberg, M., Drake, H., Zack, T., Hogmalm, J., Kooijman, E., & Åström, M. (2021).
Reconstructing craton-scale tectonic events via in situ Rb-Sr geochronology of poly-
phased vein mineralization. Terra Nova, 33(5), 502-510.
72. Tomaschek, F., Kennedy, A. K., Villa, I. M., Lagos, M., & Ballhaus, C. (2003).
Zircons from Syros, Cyclades, Greece—recrystallization and mobilization of zircon
during high-pressure metamorphism. Journal of Petrology, 44(11), 1977-2002.



73. Trotet, F., Jolivet, L., & Vidal, O. (2001). Tectono-metamorphic evolution of Syros
and Sifnos islands (Cyclades, Greece). Tectonophysics, 338(2), 179-206.
74. Rajič, K., Raimbourg, H., Gion, A. M., Lerouge, C., & Erdmann, S. (2024). Tracing
the Scale of Fluid Flow in Subduction Zone Forearcs: Implications from Fluid-Mobile
elements. Chemical Geology, 122141.
75. Redaa, A., Farkaš, J., Hassan, A., Collins, A. S., Gilbert, S., & Löhr, S. C. (2022).
Constraints from in-situ Rb-Sr dating on the timing of tectono-thermal events in the
Umm Farwah shear zone and associated Cu-Au mineralisation in the Southern
Arabian Shield, Saudi Arabia. Journal of Asian Earth Sciences, 224, 105037.
76. Taylor, A. S., & Lasaga, A. C. (1999). The role of basalt weathering in the Sr isotope
budget of the oceans. Chemical Geology, 161(1-3), 199-214.
77. Timmermann, H., Štědrá, V., Gerdes, A., Noble, S. R., Parrish, R. R., & Dörr, W.
(2004). The problem of dating high-pressure metamorphism: a U–Pb isotope and
geochemical study on eclogites and related rocks of the Mariánské Lázně Complex,
Czech Republic. Journal of Petrology, 45(7), 1311-1338.
78. Tumiati, S., Recchia, S., Remusat, L., Tiraboschi, C., Sverjensky, D. A., Manning, C.
E., ... & Poli, S. (2022). Subducted organic matter buffered by marine carbonate rules
the carbon isotopic signature of arc emissions. Nature Communications, 13(1), 2909.
79. Zack, T., & Roesel, D. (2021, December). Towards robust in-situ Rb-Sr spot ages. In
AGU Fall Meeting Abstracts (Vol. 2021, pp. V22A-04).
80. Uunk, B., Brouwer, F., ter Voorde, M., & Wijbrans, J. (2018). Understanding
phengite argon closure using single grain fusion age distributions in the Cycladic
Blueschist Unit on Syros, Greece. Earth and Planetary Science Letters, 484, 192-203.
81. Vermeesch, P. (2018). IsoplotR: A free and open toolbox for geochronology.
Geoscience Frontiers, 9(5), 1479-1493.
82. Villa. (1998). Isotopic closure. Terra nova, 10(1), 42-47.
83. Villa, I. M. (2016). Diffusion in mineral geochronometers: Present and absent.
Chemical Geology, 420, 1-10.
84. Voigt, M., Pearce, C. R., Baldermann, A., & Oelkers, E. H. (2018). Stable and
radiogenic strontium isotope fractionation during hydrothermal seawater-basalt
interaction. Geochimica et Cosmochimica Acta, 240, 131-151.
85. Volante, S., Blereau, E., Guitreau, M., Tedeschi, M., van Schijndel, V., & Cutts, K.
(2024). Current applications using key mineral phases in igneous and metamorphic
geology: perspectives for the future. Geological Society, London, Special
Publications, 537(1), 57-121.
86. Wang, C., Alard, O., Lai, Y. J., Foley, S. F., Liu, Y., Munnikhuis, J., & Wang, Y.
(2022). Advances in in-situ Rb-Sr dating using LA-ICP-MS/MS: applications to
igneous rocks of all ages and to the identification of unrecognized metamorphic
events. Chemical Geology, 610, 121073.
87. Wawrzenitz, N., Romer, R. L., Oberhänsli, R., & Dong, S. (2006). Dating of
subduction and differential exhumation of UHP rocks from the Central Dabie
Complex (E-China): constraints from microfabrics, Rb–Sr and U–Pb isotope systems.
Lithos, 89(1-2), 174-201.
88. Whitney, D. L., & Evans, B. W. (2010). Abbreviations for names of rock-forming
minerals. American mineralogist, 95(1), 185-187.
89. Wirth, E. A., Sahakian, V. J., Wallace, L. M., & Melnick, D. (2022). The occurrence
and hazards of great subduction zone earthquakes. Nature Reviews Earth &
Environment, 3(2), 125-140.
90. Zack, T., & John, T. (2007). An evaluation of reactive fluid flow and trace element
mobility in subducting slabs. Chemical Geology, 239(3-4), 199-216.



91. Zametzer, A., Kirkland, C. L., Barham, M., Hartnady, M. I., Bath, A. B., &
Rankenburg, K. (2022). Episodic alteration within a gold-bearing Archean shear zone
revealed by in situ biotite Rb–Sr dating. Precambrian Research, 382, 106872.
92. Zhao, H., Zhao, X. M., Le Roux, P. J., Zhang, W., Wang, H., Xie, L. W., ... & Yang,
Y. H. (2020). Natural clinopyroxene reference materials for in situ Sr isotopic
analysis via LA-MC-ICP-MS. Frontiers in Chemistry, 8, 594316.