# Peer review of "In-situ Rb-Sr geochronology of white mica in young metamafic and"

_Geochronology, 2024_

## Referee Comment (RC2)

[referee-annotated manuscript omitted]

---

## Author Response (AR1)

Dear Editor Prof. Daniela Rubatto

Please find below our reply to the comments from the two reviewers. We have modified the manuscript as well as added a new supplementary Figure (Figure S2; see below) in order to comply with most of the Reviewers' comments. Below you will find our responses to each of the comments and details of the corresponding manuscript sections showing how it has been modified.

Thanks for your time in considering our manuscript for publication in Geochronology.

Best regards,

Jesús Muñoz-Montecinos, on behalf of co-authors

**Reviewer #1: Dr Gyomlai**

*This study focuses on improving the in-situ Rb-Sr dating method for young metamafic and metasomatic rocks (with low Rb/Sr) by using different anchoring methods mainly based on the Sr isotopic composition of coexisting epidote. The samples used are from the well-studied area of Syros in the Cycladic Blueschist Units, and the obtained ages are used for regional interpretations. The methodological approach is innovative, at the heart of current issues in petrochronology and, therefore, of general interest to the Geochronology readership. While the obtained ages contribute to understanding the tectono-metamorphic history of Syros, there is room for improvement in the petrological and regional interpretations of these ages.*

*My main comments:*

*My first main concern is about the few petrological data to confirm that the analyzed white mica and epidote are monogenic and co-crystallized. Supplementary Fig. 4 (BSE images) is used to highlight the lack of chemical zoning patterns in white mica. However, there are high contrast differences, in particular in Fig. S4 B and F. Furthermore, the ablation spot of 80-100 µm are quite large compared to the mica grains and likely to sample these small zonings. It is possible that white mica underwent partial fluid-assisted dissolution/precipitation, in particular in a shear zone (as studied here in the Grizzas area), and I think chemical maps are necessary to see potential several mica generations (Al or Si for pressure indicator and Na if presence of paragonite; which were not measured during laser ablation). Furthermore, the compositions in major elements of both the analyzed white micas and epidotes are missing and is very important to confirm the homogeneity of the dated phases as well as placing them on the P-T path and corroborate the obtained ages (in particular for samples from Grizzas to confirm the near-peak crystallization).*

*All the images were acquired in high contrast mode and the difference in brightness (which is due to chemical zoning patterns) is very subtle, which suggest that the chemical variations between cores and thin rims are negligeable. This is confirmed by EPMA data (**see new Figure S2; attached below**) in which both cores and rims of the Grizzas micas correspond to Si-rich and high XMg white mica (phengitic) compositions that are typically found in blueschist-facies rock. Conversely, lower XMg*

*values characterise micas from the retrograde blueschist to greenschist-facies rocks at Megas Gialos (**Figure S2**). The paragonite substitution is inexistent in these micas as can be seen in **Figure S2**.*

*We believe that the relatively high contrast in mica in Figure S4B (**now Figure S1B**) provides a biased picture of the micas in this study because this mica was specifically imaged to show a zoning pattern. In Figure 2, white mica grains display a wide range of grain sizes, from mm (Figure 2B) to hundreds of μm in size (Figure 2H). We targeted the larger grains to avoid potentially retrogressed marginal parts of these grains.*

*Unfortunately, the chemical composition of the phases does not provide quantitative information about P-T conditions without applying thermodynamic modelling (see answer to the comment below). Such approach is, however, beyond the scope of this paper, specially because previous investigations have resulted in highly heterogenous and poorly constrained results ranging from 0.6 – 2.0 GPa and 400 – 600 °C as noted by the reviewer below.*

*The text addresses the homogeneity, or lack thereof, in the Sr isotope systematics of epidote in the examined samples in section "Epidote Sr isotopes". The latter case implies lack of equilibrium and clearly presents a limitation of the suggested approach.*

***We added an additional Figure to the supplementary material showing the chemical composition of white micas for most analysed samples highlighting the homogeneity in chemical composition (no significant differences between cores and rims, except for some micas in greenschist-facies rocks from Megas Gialos and the metasedimentary rock samples SYGR45).***

[Figure]

**Supplementary Figure S2.** A - F. Compositional diagrams of white mica.

*[New Figure S2: Details on the EPMA methodology are also included in the supplementary material]*

*In the discussion all the preferred ages (anchored) are used to discuss near-peak HP metasomatic event(s) at Syros but their unanchored values and the impact of the shear zone activity are not discussed. Indeed, only four samples are identified as "Metasomatic rinds, metasomatized metagabbro and veins" and only SYGR37 & SYGR38 correspond to the metasomatic rinds showing the significant change of the chemical composition of the rock which define metasomatism. The other two samples are SYGR41 a vein and SYGR42 an altered metagabbro with older unanchored ages (45 ± 11 Ma & 46 ± 9 Ma) and no epidote data. In sample SYGR38, the epidote composition is highly variable (drastically more than other samples; Fig. S1) and could indicate several generations of epidote with various fluid impulses. The unanchored isochron age is 43 ± 10 Ma. In sample SYGR37, no epidote was analyzed and the unanchored isochron age is 32.3 ± 7.5 Ma. As mentioned by the authors, metasomatism can impact the Sr composition of the system, for example with a more radiogenic signature and therefore giving younger ages. Because of this fact, of the sensibility of the ages on anchoring (Cf table S5) and of the absence of constrained metasomatic epidote composition, the interpretation of a HP metasomatic event should be more nuanced as it is possibly younger than presented in the discussion and similar to results from Gyomlai et al. (2023a), sometime inaccurately cited in the text (as detailed below). Coming back to my previous point, the chemistry of the dated phases could help constrain these ages. Indeed, studies on these metasomatic rinds suggest relatively low conditions (1.17–1.23 GPa, 500–550° C: Breeding et al., 2004; at 0.62–0.72 GPa, 400–430° C: Marschall et al., 2006; ∼2 GPa, 430°C: Miller et al., 2009 and ∼0.90–1.15 GPa and 500–600°C: Gyomlai et al., 2021) and therefore a relatively young event (⩽40 Ma).*

*The Sr isotope composition of epidote in sample SYGR38 is highly variable as noted by the reviewer, however most data points cluster to the more radiogenic end of the measured spectrum (with only four data points plotting below 0.706). In Figure S4 (**now Figure S5**) we demonstrate that by changing the epidote anchor to the "least" radiogenic signature the age gets older (46.5 ± 5.6 Ma) as expected. Yet, this age is within uncertainty of the anchored age in Figure 4B (43.0 ± 5.4 Ma) where the weighted mean ($^{87}Sr/^{86}Sr = 0.70767$) of all epidote analyses is employed. This is to say that the choice of epidote Sr isotope composition has a limited impact on the calculate Rb-Sr age for sample SYGR38. As can be seen in the thin section photomicrograph in Figure 2C, the texture of this metasomatic rind is characterized by coarse grained mica laths in sharp contact with epidote, as opposed to the host blueschist with displays a much finer grain size and a well-developed foliation. This suggests that white mica and epidote both co-crystallized during the metasomatic event(s). Chemically, epidote is known to display crystallographically controlled chemical variations and the substitution between $Fe^{3+}$ and $Al^{3+}$ may span a wide range of compositions in the blueschist-facies. Therefore, its chemistry is not helpful to constrain P-T conditions (especially in open systems) without considering thermodynamic modelling. On the other hand, we agree that the chemistry of white mica can give better hints to constrain the P-T conditions of fluid infiltration. In the new white mica chemical data added to the supplementary information, we show that this mica is Si-rich while also displaying tschemrkak substitution, high XMg and low Na. The variation of white mica composition is very limited, e.g., Si of between 3.37 to 3.41 a.p.f.u. Such high Si values are typically found in high pressure blueschist-facies rocks, and therefore all the white mica crystals, including cores and rims, crystallized during HP metasomatism. This contrasts with other white mica grains from samples from Megas Gialos and the rims of micas in the metasedimentary sample SYGR45 which display higher muscovitic component (**new Figure S2**).*

*In the new version of the manuscript, we acknowledge the previous studies addressing the P-T conditions of metasomatic rocks in Syros but highlight that these results are variable and hence do not allow to robustly constrain the P-T conditions. They rather suggest that these lithologies can form along the whole PT range from near-peak to early exhumation (lines L731-739):*

*"This enables us to constrain localized shear zone activity under HP conditions within the subduction channel in the presence of fluids. ==Although previous studies have attempted to estimate the P-T==*

*conditions of formation of these metasomatic lithologies along the Kampos belt, the results vary widely potentially suggesting that metasomatism may have occurred throughout the prograde to exhumation path (Marschall et al., 2006; Miller et al., 2009; Gyomlai et al., 2021). This is confirmed by novel reaction-path thermodynamic modelling approaches, demonstrating that bulk rock compositions, particularly the activity of elements such as Ca and Mg, play a primary role in the formation of these metasomatic rocks (Codillo et al., 2022)."*

The studied rocks from Grizzas are from a shear zone, and likely impacted by fluid-assisted dissolution/precipitation and not necessarily metasomatism (with a significant change of the chemical composition of the rock). The obtain ages should therefore be discuss as such and compared with previous ages from shear zones and in particular with ages from Laurent et al. (2021) which dated the continuous activity of the Lia shear zone (superior limit of the Kampos-Lia unit in which the studied Grizzas shear zone is) from 51 to 24 Ma.

*Fluid-assisted dissolution/precipitation involving chemical changes is equivalent to metasomatism. We have added a sentence acknowledging the results from Laurent et al. (2021) at L721-729:*

*"Thus, our data suggest that metasomatism began under near-peak metamorphic conditions and continued during the early stages of HP exhumation. These results agree with Ar-Ar ages constraining the activity of the Lia shear zone (norther boundary of the Kampos belt; see Figure 1A) at near-peak to blueschist-facies exhumation conditions in the ~51 to 35 Ma range (and locally down to 23 Ma due to later greenschist-facies activity; Laurent et al., 2021). Furthermore, Barnes et al. (2024) reported an in-situ white mica Rb-Sr age of 44.5 ± 3.1 Ma for a metasomatic eclogite (Delfini locality; presumably Middle CUB), suggesting that metasomatism in this section of the nappe stack also initiated at HP conditions."*

Other comments/questions:

l. 43: what do you mean by 'nutrient' ?

*We refer to "nutrient" to organic compounds being subducted. For the sake of simplicity, we have deleted this word from the manuscript.*

l. 44: add e.g., before Breeding et al., 2004

*Text modified accordingly.*

l. 57: or with a low concentration in U

*Text modified at L59.*

l. 60-62: If the U-bearing minerals crystallize during mid- to low-temperature metamorphic and metasomatic events, a higher closure temperature would not be an issue to date them. In Syros a lot of potential U-bearing minerals are crystallizing during metasomatism (i.e., apatite or rutile) but their concentrations in U are usually too low for dating.

*This is a good point. We have deleted this sentence from the text in order to shorten the introduction, but added a sentence acknowledging the low concentration of U in these phases (L59): "…U-bearing accessory phases such as zircon, allanite, titanite, rutile, and apatite which may be scarce, too small to be targeted, or have low-U concentrations (e.g., Timmermann et al., 2004; Rubatto et al., 2011; Regis et al., 2014; Engi et al. 2017; Holtmann et al., 2022; Volante et al. 2024 and references therein)."*

*l. 77-83: a main argument should be the coexistence of several generation of micas in one sample.*

*Thanks for this. Text modified accordingly (L75-79):* "These include: i) Sr isotope disequilibrium between micas and the other mineral phases; ii) coexistence of several generations of micas; iii) post-deformation, low-temperature magmatic alteration or fluid-assisted recrystallization; iv) thermally-induced diffusion processes (Glodny and Ring 2022); and v) potential inheritance within mica grains or across mica populations (Villa, 2016; Barnes et al., 2024)."

*l. 114-115: "However, their large uncertainties precluded the distinction between peak-pressure metamorphism, retrogression and/or partial recrystallisation of white mica under blueschist- to greenschist-facies conditions.": In Gyomlai et al. (2023a), several samples (with potentially or non-metasomatic micas) have indeed high uncertainty but the dating of neo-crystallized mica, formed during fluid-rock interactions, have relatively low uncertainties (transect L: 36.3 ± 5.1 Ma; 36.1 ± 4.7 Ma; 36.3 ± 3.1 Ma; sample near Lia beach: 12.0 ± 3.1 Ma) and link to neo-crystallization of mica during retrogression thanks to potassium brought by the fluid.*

*In order to shorten the introduction section, as suggested by the second reviewer Dr Ribeiro, we have deleted these sentences from the manuscript.*

*l. 141-142: "Although the general architecture and structural relationships of blueschist- to eclogite-facies rocks in Syros are still debated (e.g., Keiter et al., 2011; Laurent et al., 2018; Kotowski et al., 2022), the subdivision of geological units, P-T conditions and the timing of metamorphic burial and exhumation are well-constrained, making Syros an ideal case study for our purpose.": As you say, the architecture is still debated but so is the subdivision of geological units (e.g., Laurent et al., 2018 vs. Kotowski et al., 2022), P-T conditions (e.g., Cisneros et al., 2021 vs. Gorce et al., 2021) and the timing of metamorphic burial (e.g., Laurent et al., 2018 vs. Kotowski et al., 2022).*

*We partially agree. It is accepted that peak metamorphic conditions were reached at around 1.6 – 2 GPa (e.g., Laurent et al., 2018; Gorce et al., 2013; Behr et al., 2018; Spear et al., 2024) at 50 to 55 Ma, followed by blueschist facies exhumation at 50 to 40 Ma (Glodny and Ring, 2022) and subsequent greenschist facies at ages younger than 40 Ma. On the other hand, the P-T conditions of metasomatism remain a topic of debate, as highlighted by Dr. Gyomlai. This paper, however, provides new insights into the timing of metasomatism and its possible connection to some of the previously identified tectonometamorphic events.*

*l. 182: Grizzas instead of Gryzza*

*Text corrected accordingly.*

*l. 194: similar to recent age of Tual et al., 2022*

*The recent age obtained by Lu-Hf geochronology of garnet was added in the text (L170-172):* "Similar peak ages of 51.8 ± 0.1 Ma were obtained by Lu-Hf geochronology of garnet in a metasedimentary rock from the Fabrikas outcrop in south Syros Island (Tual et a., 2022)."

*l. 204-206: -"Gyomlai et al. (2023a) obtained in-situ mica Rb-Sr ages from a single, c. 2 m-thick outcrop in the range of 52.5 ± 11.6 to 12 ± 3.1 Ma, inferred to date metasomatism of metamafic rocks during HP and fluid-rock interaction during late exhumation, respectively.": Only samples n57, n10 and n1 are from this 2m-thick outcrop (Fig. 2a) and give a constrained 36 Ma metasomatic event. We consider this event as the 'main' one because of the observations at the scale of the unit (Gyomlai et al., 2021) but we are convinced of the presence of other fluid-rock interaction events. The other samples are from other parts of the ophiolitic unit (Fig. 1).*

*We now make clear that samples N57, N10 and n1 belong to this transect, while the other samples are from other locations within the belt (L176-181):* "*Gyomlai et al. (2023a) obtained* three in-situ mica Rb-Sr ages from an outcrop within the Kampos belt (Lia side) in the range of 36.3 ± 5.1 Ma to 36.1 ± 4.7 Ma, inferred to date metasomatism of metamafic rocks during blueschist- to greenschist-facies exhumation. The authors also reported older ages in the range of 52.5 ± 11.6 to 39.8 ± 7.4 Ma (Kampos belt, Lia side), but it is unclear whether these ages represent metasomatism and/or mineral (re)crystallization during peak metamorphism or retrogression during HP to late exhumation."

*l. 224-225: What is the extent and width, of the Grizzas shear zone?*

*The Grizzas shear zone is a complex shear zone that cannot be clearly traced inland, challenging the assess of its extent and width. This is also the case for many other structures in the Kampos Belt, where blocks of metagabbros are embedded in metasomatized rocks defining shear zones (including those studied by Gyomlai and coworkers).*

*l. 250 & 254 & 260 & 304: Figure S4 instead of S1.*

*We have revised and corrected the organization of the figures throughout the entire text.*

*Methods: In supplementary it seems you used BCR2G and BHVO2G as secondary standard? Precise it in the methods and how well their values are reproduced. Out of curiosity: if you calculate ages with BCR2G as a primary standard do you obtain similar results?*

*We have added the following statement (L374-376):* "Natural glass standards BCR-2G and BHVO-2G were also analysed as a quality measure of the Sr isotope analyses and returned values broadly consistent with accepted values (**Supplementary Table S2**)."

*We did not attempt to calculate mica Rb/Sr ages using BCR-2G because silicate glasses are well known to have different down-hole fractionation behavior for Rb and Sr compared to micas (e.g., Redaa et al., 2021 JAAS; Huang et al., 2023 GGR), hence producing potentially spurious ages.*

*l. 427-439: add MSWD values for the weighted means*

*We have included all MSWD values in Figure S3.*

*l. 467-468: "For Grizzas, employing this value is justified by the fact that the weighted mean of epidote Sr isotopes are ~0.708 for three or the four analysed samples": For sample SYGR50, it seems more accurate to anchor its isochron to its epidote Sr isotopes of ~0.705. In an open system such as this fluid rich shear zone, it would be logical to have different Sr composition through time even if the samples are close.*

*No age is reported for sample SYGR50 as only epidote was measured from this sample (L 189-190).*

*l: 479: here and in the rest of the text, uncertainties for ages should be 2σ and not 2SE. Better to use "σ" instead of "s" throughout the article.*

*Considering how IsoplotR treats the age uncertainties, we believe that 2SE is a more transparent way to report them. 2s is equivalent to 95% of confidence, which is not necessarily the case with IsoplotR.*

*Fig. 4: Add the rock type corresponding to each sample. For SYGR58, where are the 36 [mica Rb-Sr isotope] unconsidered data?*

*Table 1 summarizes all the investigated samples, where the rock-type is listed in column 2.*

*All the mica Rb-Sr isotope data are in Supplementary Table S3. The analyses reported in red are not considered in the isochronous regressions because they contain low Rb and probably do not represent mica grains.*

Table 2. Add MSWD for weighted mean, 1 or 2σ for y-intercept and 2σ uncertainties instead of 2SE for ages.

*We don't believe it is necessary to report MSWD values for the weighted means of epidote Sr isotope data because the 2SE provides a clear picture of data spread and the table is already quite busy. All the age uncertainties represent 2SE not 2s.*

l. 595-596: "Low 87Sr contents are associated with large uncertainties for 87Sr/86Sr, which systematically exceed 1% (2SE) for individual measurements (Figure 5A)": not really illustrated as Figure 5A is the "comparison of relative standard deviations (1 SE, standard error) of unanchored mica Rb/Sr ages and average 87Sr/86Sr uncertainties".

*We have replaced "Figure 5A" with "Supplementary Table S3". The statement that "Low $^{87}$Sr contents are associated with large uncertainties for $^{87}$Sr/$^{86}$Sr" is intuitive and does not need a visual representation.*

Figure 5: " relative standard deviations (1 SE, standard error)", if a relative standard deviation, then it is RSD.

*Amended to "relative standard error (RSE)".*

l. 602: very few spots for this inverse correlation, which is not very clear, you could add the literature data.

*We have considered adding our own data from other studies to compare results obtained using similar instrumental conditions and analytical protocol. However, the relative uncertainty of mica Rb-Sr ages is not just a function of Rb-Sr spread and uncertainty of Rb-Sr and $^{87}$Sr/$^{86}$Sr, but also other factors including the absolute age which affects ingrowth of radiogenic $^{87}$Sr. Therefore, we have opted to keep our focus on the (relatively young) Syros samples. We should add it is not the purpose of Figure 5 to demonstrate a robust correlation between relative $^{87}$Rb/$^{86}$Sr spread and age uncertainty – this is what the Syros data show.*

l. 613: "(and probably chemical equilibrium)": then it would be interesting to measure the composition in major elements of the epidote and mica.

*Measuring the composition of epidote and mica does not ensure chemical equilibrium. The composition of epidote in blueschist-facies rocks can span the whole range between pistacite to zoisite/clinozoisite depending on the amount of Fe and the oxygen fugacity. In the revised version of the manuscript we have included compositional data for white mica (see previous replies).*

Figure 6: It would be important to illustrate here the accuracy of ages (with the MSWD), for example with a colorbar or in the y-axis of Fig. 6A. Furthermore, I think adding anchored ages for all samples at both 0.7080 and 0.7050 would help the reader understand the impact of anchoring.

*The symbols are already colour-coded to illustrate the anchoring approach, or lack thereof, and adding details of MSWD would not enhance the comparative scope of this figure, but rather make it more complicated. Regardless, the MSWD values of anchored and non-anchored regressions are remarkably similar – see more details below including an update of originally incorrected MSWD values for some anchored regressions.*

*The mica Rb/Sr isochrons are currently anchored to the epidote Sr isotope value (where available) and to a representative Sr isotope composition, which is based on the Sr isotope systematics of the Syros*

*rocks. We don't think that anchoring to values that are not necessarily representative of the initial Sr isotope composition in that sample adds value to this figure. The comparison suggested by Dr Gyomlai is already included in Supplementary Table S5 where the mica Rb/Sr isochrons are anchored to a range of $^{87}Sr/^{86}Sr$ values.*

l. 683-696: *Here you should discuss with the overall ages from the literature and not only the Rb-Sr ages.*

*The discussion includes ages from different methods. The compilation we present in Figure 8 is a synthesis of Zircon (U-Pb), Garnet (Lu-Hf) and white mica Ar-Ar and Rb-Sr from the literature compiled by Glodny and Ring (2022) and Kotowski et al. (2022). This information, which is already included in the discussion, has been added to the caption of Figure 8 (L764).*

l.728-729: *"however, the significantly larger uncertainties of their mica Rb/Sr ages for similar rock types should be noted" Indeed uncertainties for several samples are high, but samples dating metasomatic events (36.3 ± 5.1 Ma; 36.1 ± 4.7 Ma; 36.3 ± 3.1 Ma; 12.0 ± 3.1 Ma) have similar uncertainties to your anchored ages with epidote and better accuracy.*

*Gyomlai et al. (2023a) reported nine ages from which six have large uncertainties (39.8 +/- 7.4; 49.5 +/- 9.5; 12.0 +/- 3.1 Ma; 52.5 +/- 11.6; 46.7 +/- 8.5 Ma), which is to what we referred to in our statement. In the new version of the manuscript we have deleted the sentence "however, the significantly larger uncertainties of their mica Rb-Sr ages for similar rock types should be noted".*

l. 725-738: *"with previous interpretations which suggested that metasomatism along the entire Kampos Belt occurred as a discrete pulse during the latest stages of exhumation (Gyomlai et al., 2023a)". We discuss that fluid rock interactions along the subduction interface occur as several discrete pulses which may be very local (i.e., the 12 Ma event). Citing Gyomlai et al. (2023a): "importance of identifying other potential local metasomatic events in the Kampos Unit to constrain the extent of fluid–rock interaction during subduction and exhumation.". Your data are in agreement with that, showing one or several pulsated near-peak fluid ingression(s). This event(s) was ever not recorded/preserved in our samples or not occurring in the area we sampled. This event may be compatible, as you mentioned, with our sample 9a (with high uncertainties) but as we discussed we have no clear petrographic evidence as to whether white mica in samples 9a (as well as 9c and 27) is purely metasomatic or metamorphic relict. Furthermore, your age for sample SYGR58 (36.1 ± 2.1 Ma) is highly consistent with our results.*

*The purpose of our statement is to clarify that our results disagree with the occurrence of a "main" metasomatic pulse at 36 Ma, as suggested by Gyomlai et al. (2023a), but rather supports continuous metasomatism starting at near-peak conditions or during the earliest exhumation stages and continuing during exhumation. In our study shear zone samples and metasomatic rocks yield **new ages** dating metasomatism from **46.6 to 41.1 Ma**, which are comparable to either near-peak metamorphism or the earliest stages of exhumation under blueschist-facies conditions (rather than the transition to greenschists facies during latter exhumation) and the mineral assemblages also support HP during metasomatism. In addition, the recent paper from Barnes et al. (2024) examined metasomatic eclogite assemblages from a locality that belongs to the underlying nappe ("Middle-CBU" in the sense of Glodny and Ring, 2022) and yielded an in situ mica Rb-Sr age of 44.5 +/- 3.1 that is consistent with our results. Similarly, Gyomlai et al. (2023a) detected potentially older metasomatic ages from blackwalls and rinds (52.5 +/- 11.6 Ma and 49.5 +/-9.5 Ma) around metagabbros (in agreement with our results), which likely date similar metasomatism at near peak metamorphic conditions as the one identified in our study.*

*We have modified the manuscript deleting the sentence "Thus, our data is at odds with previous interpretations which suggested that metasomatism along the entire Kampos Belt occurred as discrete*

*pulses during the latest stages of exhumation (Gyomlai et al., 2023a)." and added some clarifications at L718-729:*

"Our data points to at least one event of HP metasomatism ==and fluid-rock interactions== along ==the Grizzas shear zone== within the range of ==46.6 ± 4.6== Ma to 41.1 ± 3.1 Ma. Due to method uncertainties, distinguishing between multiple events within this time range is not feasible. ==Thus, our data suggest that fluid-rock interactions and metasomatism began under near-peak metamorphic conditions and continued during the early stages of HP exhumation.== These results agree with Ar-Ar ages constraining the activity of the Lia shear zone (norther boundary of the Kampos belt; see Figure 1A) at near-peak to blueschist-facies exhumation conditions in the ~51 to 35 Ma range (and locally down to 23 Ma due to later greenschist-facies activity; Laurent et al., 2021). Furthermore, Barnes et al. (2024) reported an in-situ white mica Rb-Sr age of 44.5 ± 3.1 Ma for a metasomatic eclogite (Delfini locality; presumably Middle CUB), ==suggesting that metasomatism in this section of the nappe stack also initiated at HP conditions.==

,,

*l. 732: "Our data points to at least one event of HP metasomatism within the range of 47.2 ± 3.8 Ma to 41.1 ± 3.1 Ma" Why considering all samples here and not only the metasomatic ones (SYGR37 & SYGR38, ± SYGR41, SYGR42) ? I think ages should be discussed as both preferred anchored and unanchored values, as anchoring with epidote increase (sometime drastically) the MSWD of the ages.*

*To address the age of metasomatism we agree in that we should focus on the metasomatic rocks rather than considering the whole sample set. In the previous version of the manuscript, we incorrectly considered one pristine, "non"moetasomatic sample (SYGR45 = 47.2 ± 3.8 Ma) as a metasomatic sample; we corrected this in the new version of the manuscript and discuss about the ages in the range of 46.6 +/- 4.6 (upper range for sample SYGR42) to 41.1 +/- 3.1 (lower range for sample SYGR37). In addition, we have modified Figure 8 to enclose only the metasomatic samples (SYGR37; SYGR38; SYGR41; SYGR42) with the label "Grizzas shear zone metasomatism". Furthermore, we now plot the age of c. 47.2 Ma for the metasedimentary sample SYGR45, which is the anchored age, instead of c. 45 Ma (unanchored age).*

*Anchoring does not drastically change the MSWD values of the regressions through the mica Rb-Sr isotope data. In the original manuscript we incorrectly reported the MSWD values for 5 samples, including a very high value of >300 for sample SYGR58. The problem with the previous isochrons was that we did not 'anchor' the mica Rb-Sr isotope data, but rather added all the epidote data points to the mica-based isochronous array. Although the calculated ages do not change, the previous approach resulted in extremely high MSWD values in the two samples (and especially in SYGR58) where the epidote Sr isotope data are variable.*

*We have modified Figure 4, Supplementary Figure S2 **(now Figure S3)** and Table 2 to include the correct MSWD values and amended the text accordingly. The new MSWD values of the epidote-anchored isochronous are effectively indistinguishable from those of the non-anchored regressions, which strengthen the anchoring approach suggested in this paper.*

*l. 735: Comparing to data from Barnes et al. (2024) is difficult and should be further discussed/nuanced if it is from another tectonic unit (middle CBU) with a different P-T-d-f history (e.g., Laurent et al., 2018; Kotowski et al., 2022). Or do you assume it is from the top CBU unit as represented in figure 8?*

*No, we do not assume the samples from Barnes et al. (2024) are from the same unit. Both units (middle and top CBU) share a distinct history but in the structural pile the middle CBU represents the younging of the stack downwards (south) (Glodny and Ring, 2022; Kotowski et al., 2022). Barnes et al. (2024) presented data for two eclogite samples (along with one mica schist and a blueschist), including a metasomatic rock belonging to the Middle CBU and a "fresh" eclogite presumably belonging to the top CBU (as labelled in our Figure 8). The ages obtained by Barnes et al. (2024), in particular that for eclogite-facies metasomatism, is only used here to illustrate that even in other nappes of Syros, metasomatism also begun at near-peak conditions.*

*We agree in that the text could have been misleading in this regard, and we have modified the paragraph to make it clear that the results from Barnes et al. (2024) do not necessarily apply for our study case, but elsewhere along Syros in the context of deep metasomatic processes (lines 721-729): "Thus, our data suggest that fluid-rock interactions and metasomatism began under near-peak metamorphic conditions and continued during the early stages of HP exhumation. These results agree with Ar-Ar ages constraining the activity of the Lia shear zone (norther boundary of the Kampos belt; see Figure 1A) at near-peak to blueschist-facies exhumation conditions in the ~51 to 35 Ma range (and locally down to 23 Ma due to later greenschist-facies activity; Laurent et al., 2021). Furthermore, Barnes et al. (2024) reported an in-situ white mica Rb-Sr age of 44.5 ± 3.1 Ma for a metasomatic eclogite (Delfini locality; presumably Middle CUB), suggesting that metasomatism in this section of the nappe stack also initiated at HP conditions."*

*"*

Fig. S1 show a relatively wide dispersion of epidote 87Sr/86Sr, the MSWD of the weighted mean should be added on this figure and when anchoring the uncertainty should take into account this high MSWD (as calculated in the isoplotR software and including the square root of the MSWD).

*We have added MSWD values to Figure S1 (**new Figure S3**). The MSWD value of the weighted mean of the epidote Sr isotopes cannot be included in the calculation of isochronous regressions using IsoplotR.*

Fig. S4: In E and C, the contact between white mica and Epidote is quite irregular, is there more argument they are co-stable? Did you try to date SYGR44 and SYGR50?

*We disagree. The contact is sharp and straight. There are regions of mechanical fragmentation owing to the polishing (see attached figures). In the case of sample SYMG08.3, for example, this is a dilatational vein without evidence of post entrapment deformation, as evidenced by the fibrous habitus of epidote together with white mica laths in close contact.*

*Samples SYGR44 and SYGR50 were not dated, but Sr isotope ratios were measured in epidote to constrain the isotopic signature of Sr in mafic rocks and detect potential isotopic modifications associated with their metasomatized counterparts.*

[Figure]

*Table S2 & S3: simplify and clarify to help the reader*

*We have removed some columns and modified the number of decimal digits to make the tables easier to read.*

*Best regards,*

*Thomas Gyomlai*

**Reviewer #2: Dr Ribeiro**

*Dear Daniela Rubatto and authors,*

*I appreciate the opportunity to review this manuscript which focus on the improvements of white-mica Rb-Sr dates by anchoring the isochrons to epidote initial $^{87}Sr/^{86}Sr$. Below I provide some comments and questions that I believe could help the authors to improve their manuscript.*

*Introduction*

*I believe the introduction is well written and clearly state the issue to be addressed. Yet, I believe it is far too long and could be easily shortened and focused on the geochronology of HP-LT rocks. In the present stage, the introduction has seven paragraphs which I think is too long. Shorter and focused texts are more impactful and gets the reader attention easier. I believe the introduction would benefit from excluding the parts in which the authors present the findings of specific papers (e.g., lines 84-89; 112-119).*

*Thanks for this suggestion. In the new version of the manuscript we have shortened the introduction section by deleting the parts suggested by Dr Ribeiro, as well as shortened the overall introduction. We have reduced its length from 7 to 6 paragraphs.*

*Samples and Petrography*

*This section is well presented and Figure 2 display great photomicrographs of the samples of interest. Beautiful micas! I noticed that Table 1 does not have a caption (at least not embedded in the text along with the table). Additionally, the mineral abbreviations are not specified. Does it follow any reference? I noticed the Figure 2 caption states the mineral abbreviations from Whitney and Evans (2010), so I recommend moving this reference to Table 1 caption since it appears first in the manuscript. Additional minor comments are embedded in the text.*

*Thanks for noticing it. We agree with this comment and we have modified Table 1 by adding a caption and a reference to Whitney and Evans (2010) for mineral abbreviations.*

*Methods*

*LA-MC-ICP-MS for Sr/Sr analysis*

*I noticed the authors stated they ran the in situ Sr/Sr analysis with distinct laser fluence (4 vs. 2.5 J.cm-2) and repetition rate (5 vs. 10 Hz). I wonder why the conditions were so different, and if they found that one setting is more suitable than the other. I imagine that the higher repetition rate would increase the count rates, therefore improving internal precision. Anyway, some explanation for the use of distinct setting would be appropriate.*

*Over the last few years we have been experimenting different settings and recently adopted a higher ablation frequency (10 Hz) following the observation of Huang et al. (2023 GGR), who found lower discrepancy between in-situ mica Rb/Sr ages and independent age constraints using high ablation frequency and large spot size. However, to maintain the quantification of $^{86}Sr$ (as $^{86}Sr^{16}O$) in NIST610 in pulse counting mode – as it is the case for mica –we had to apply a lower energy fluence. In any case, we did not observe any systematic variation in the accuracy of the secondary mica standards using the two analytical setups as it can be noticed by the age results reported in Supplementary Table S2.*

*Sr/Sr data reduction in iolite 4 – Did the authors write their own data reduction scheme? If so, please refer to "in-house" data reduction scheme, or cite the original reference for the scheme. If this is an "in-house", I recommend the authors to include in the manuscript and make it available to the community. I believe this would increase the manuscript impact, garnering more citation through time besides helping the community to continue developing such technique.*

*The Sr isotope data reduction scheme is embedded in the Iolite 4 software and, while one of the authors (Giuliani) provided input to the software developer and tested a beta version of the package, we do not believe a specific reference should be included. The method builds upon the pioneering work of Woodhead, Paton and co-workers at the University of Melbourne, which is properly cited in the manuscript (Paton et al., 2007 GGR; Paton et al., 2011 JAAS).*

*Reference materials – I find interesting that the authors only report the use of cpx reference materials, without considering the commonly used NIST and other glasses. Is there any reason for that? Are the glasses not suitable to check instrument bias, sensitivity etc…? I supposed the tuning was carried out using NIST glasses? I would be good if the authors could address these queries.*

*Correct, initial tuning is done using NIST610 for optimal Sr signal followed by further tuning using a fragment of modern marine carbonate for optimal peak shape. All the details are in previous publications which are cited in the text (Fitzpayne et al., 2023 Chem Geol; Pimenta Silva et al., 2023 Contr Mineral Petrol) and are not repeated here.*

*Additional glasses were not measured for Sr isotopes by LA-MC-ICP-MS as secondary standards because they commonly contain Rb and hence are not ideal targets for Sr isotope analysis – although low Rb/Sr glasses can certainly be analyzed. As at the time of this project we did not have any epidote standard for Sr isotope determination, we opted to employ clinopyroxene as a proxy – while recognizing the different matrix.*

*LA-ICP-MS/MS for Rb-Sr analysis*

*The authors stated that the in situ Rb-Sr analyses were carried out following the method outlined in Giuliani et al. (2024) and Ceccato et al. (2024). Although I acknowledge that these papers described the settings used in the ETH lab, they do not represent the main reference for the methodology which needs to be acknowledged in the manuscript in my opinion. The pioneering work of Zack and Hogmalm (2016) and Hogmalm et al. (2017) are the appropriate references that describe the technique itself. Thus, I suggest the authors to modify the text to indicate that the method applied follow these two papers, with instrumental settings following Giuliani et al. (2024) and Ceccato et al. (2024).*

*We have included references to Zack and Hogmalm (2016 Chem Geol) and Hogmalm et al. (2017 JAAS) in the methods section.*

*The authors used NIST612 for single-quad tuning, and then shifted to NIST610 for MS/MS mode. Why? Just simpler to use NIST610 for both stages.*

*We have been zapping NIST612 for initial tuning since the early days of this effort because NIST612 has much lower (one order of magnitude) concentrations of trace elements than NIST610 hence preventing a flood of unnecessary ions to the detector.*

*There is no additional information about error propagation into the Rb/Sr and Sr/Sr ratios, which needs to be considered to include the internal and external variability/uncertainties. I recommend the authors to mention this in their manuscript. In case they didn't propagate uncertainties, I strongly recommend doing it otherwise the data do not capture all natural and instrumental variability.*

*All the uncertainties have been fully propagated by quadratic addition as specified by Giuliani et al. (2024 Chem Geol) where additional details of the methodology are included. Some of that information is not repeated here where the focus is on the application of mica Rb/Sr dating rather than on the method development, which was the focus of our previous contribution (Giuliani et al., 2024).*

There is also no specification regarding the isochron calculations, including Rb decay constant and software (IsoplotR I suspect?). Please include such information and the level of uncertainty in the plots.

*We have included this information (IsoplotR and 87Rb decay constant plus relevant references) at L395-396. IsoplotR was already mentioned in all the figure captions.*

Results

Epidote Sr

There is a bit of repetition in the first paragraph when mentioning the number of samples analysed etc. This could be simplified.

*We agree in that this could sound a bit repetitive, however, we believe that it is necessary as this explanation is given earlier in the beginning of the petrography section and the reader could benefit from this "reminder".*

Figure 3 does not include error bars for individual analysis. Is this due to very small uncertainties (smaller than the symbol)? If the uncertainties are visible in the diagram, I recommend adding them to the figure. Otherwise, please specify in the figure caption that "uncertainties are smaller than the symbol size".

*Correct, the uncertainties are very small (see Table S1). We have added "uncertainties are smaller than the symbol size" in the caption in Figure 3.*

White mica Rb-Sr

Sample SYGR36 yield a certainly more precise isochron age when anchored to the epidote Sr/Sr ratio, however the MSWD gets incredibly high (4.2 compared to 0.9 from the unanchored isochron). I believe this will be further discussed in the manuscript, however I recommend the authors to acknowledge the increase of such important statistical parameters. Additionally, it seems that the authors used a 1 SE uncertainty for the Sr/Sr ratio, which is not appropriate when linking it to 2SE uncertainties from the Rb-Sr analysis. Keep it all consistent with 2 SE for all ratios. The same comments apply to the result description from all samples, and further discussions.

*We wish to thank Dr Ribeiro (as well as Dr Gyomlai) for spotting this inconsistency in the MSWD values which has prompted us to double-check all the regressions and spot a mistake in the original calculations. As noted in our replies to Dr Gyomlai, the MSWD values of the epidote-anchored isochronous arrays were incorrectly reported in the original version of this manuscript. We have recalculated all the mica Rb-Sr ages using the correct weighted mean values of epidote 87Sr/86Sr and amended text, Figures 4 and S2 (now Figure S4), and table 2 accordingly. The difference of MSWD between epidote-anchored and non-anchored regressions through the mica Rb-Sr data is now almost indistinguishable.*

*2SE have been consistently used for the weighted mean of the epidote Sr isotope ratios. Using 1SE has no noticeable effect based on additional testing undertaken during manuscript revision.*

*Discussion*

*In lines 604-608, the authors mentioned that the unanchored Rb-Sr isochrons were imprecise (which I obviously agree) and inaccurate. The accuracy of the isochron age is difficult to assess, as it would require proper ID solution to compared with absolute age. However, as I mentioned before, the anchoring approach increased the MSWD from 0.9 to 4.2, which clearly points to a overdispersed data and statistically meaningless. Therefore, the statistical parameters do not support the use of Sr/Sr ratio to anchor this isochron, as it might be in chemical disequilibrium with the Rb-Sr systematics in micas. I recommend the authors to address this in the manuscript.*

*As noted above, the MSWD values of the epidote-anchored regressions were incorrectly reported. After amendment, there is a negligible difference between the MSWD values of epidote-anchored and not-anchored mica Rb-Sr ages.*

*In lines 620-621, the authors mentioned the precision improvement when anchoring the Rb-Sr data with Sr/Sr ratios obtained with MC-ICP-MS. Although I agree that such approach is reasonable, the outcome is rather expected as mentioned by the authors (lines 634-635). The isochron uncertainty will be largely controlled by the incredibly precise Sr/Sr ratio compared to the much larger white mica Rb-Sr internal uncertainty. We can see this as a mixing line between a very precise ratio with less precise data. Another appropriate solution that might be interesting to highlight is the use of new series of multicollector instruments equipped with reaction cells (Neoma from Thermo, and Sapphire from NuPlasma). The Neoma, for instance, was designed to analyse samples with low Rb concentrations increasing the internal precision.*

*Anchoring mica Rb-Sr isotope data is beneficial to get better age constraints regardless of the instrument employed to obtain the mica Rb-Sr data – a collision-cell multi-collector instrument such as the Neoma being certainly advantageous. However, in collision-mode the Sr counting statistics of a multi-collector instrument are expected to decrease compared to Sr isotope analyses without gas in the collision cell – or using an MC-ICPMS without a collision cell. We believe this discussion goes beyond the boundaries of our contribution.*

*Summary*

*The authors highlight that "By anchoring these data to a low Rb/Sr phase such as epidote, age precision improved by up to six times", which is clearly true. But I think it is reasonable to address that this is an "analytical bias" given that micas and epidotes were analysed under distinct instrumental resolution. For example, I don't know if this higher isochron date precision would be significantly increased if low-Rb phases were also measured with a triple quadrupole along with micas. My personal experience shows that yes – it does improve a bit, but such significant improvement demonstrated in this manuscript is strongly controlled by the distinct techniques used. I think the take-message is cool – we should use low-Rb phases to anchor the isochrons, especially for very young samples! But the instrumental part could also be acknowledged.*

*We fully agree with Dr Ribeiro and have added a relevant statement in the Conclusions (L779-782):* ==*"Such improvement is contingent to the employment of a MC-ICP-MS instrument to obtain accurate and precise Sr isotope values for the low Rb/Sr phase by laser ablation compared to the considerably lower precision of similar analyses by LA-ICP-MS/MS (Barnes et al., 2024)."*==

*Unfortunately I don't have experience in the Syros geological setting, so I hope other reviewers could better assessed this section.*

*I hope my comments and suggestions assist the authors to improve and amend some parts of the manuscript. I really liked the dataset and I hope it sees the light of the day.*

*Kind regards,*

*Bruno Ribeiro*

**Reviewer #3: Prof. Dr. D. Rubatto**

The revision is detailed and addresses most of the reviewers' concerns with the addition of new data on the chemistry of mica (Fig. S2) to better represent the presence or absence of major zonation, clarification of the methodology (reference material, evaluation procedure and uncertainty propagation), modification of the MSWD values presented and their discussion. References to previous work have been added where relevant, the text has been modified to clarify language and some concepts, order and reference of figures have been reviewed and corrected.

Additional private note (visible to authors and reviewers only):

Dear Dr Muñoz-Montecinos and co-authors, thank you for the careful revision, which I am happy to accept.

I agree with Gyomlai that the word metasomatism should be used with caution, fluid-rock interaction causing mineral recrystallisation is not necessarily metasomatism if the fluid is aqueous and the rock composition does not change substantially in major elements. This is best defined as fluid-rock interaction (hydration/dehydration is generally not considered metasomatism), metasomatism should only be used where there is evidence of a significant change in bulk rock major element composition.

I agree with Ribeiro that the intro is a bit wordy and not very focused. I add a few comments for shortening and clarification

Lines 55-67. It is not very meaningful to compare with U-Pb, which targets effusive things. Just state the advantages of Rb-Sr in general. Allanite U-Pb could also be mentioned as it is commonly found in HP rocks (e.g. Regis et al. 2014 and Rubatto et al. 2011). Also, Rb-Sr has always been present and has been used extensively in the past, so it is not a new exploration.

69: Mica is only common in AOC and metasediments, not in dry eclogites.

73: Static fluid absent is a very unlikely condition in a dehydrating slab, maybe this needs to be clarified.

*We sincerely appreciate the insightful feedback and constructive comments provided by the Editor, Prof. Dr. D. Rubatto.*

*We agree with all the suggestions and have revised the manuscript accordingly, as detailed below:*

1) *The introduction has been further shortened and revised to clarify the focus of this work. References to effusive processes have been removed, while references related to allanite dating have been added. The section on 'new exploration' has been deleted, and we have clarified that white mica is typically found only in altered oceanic crust and metasediments. Additionally, we*

*have addressed the issue concerning static and fluid-absent conditions. Please see the revised introduction in lines 44 to 121.*

2) *We have taken care in using the term 'metasomatism' and have also included the phrase 'fluid-rock interactions' where appropriate. Additionally, we have revised Figure 8 to refer to the 'Grizzas shear zone metasomatism and fluid-rock interactions' instead of using 'metasomatism' alone (see below).*

[Figure]